# NOT ALL PROMPTS ARE MADE EQUAL: PROMPT-BASED PRUNING OF TEXT-TO-IMAGE DIFFUSION MODELS

**Alireza Ganjdanesh**[*1] , **Reza Shirkavand**[*1] , **Shangqian Gao** [2], **Heng Huang** [1†]

[1]Department of Computer Science, University of Maryland, College Park
[2]Department of Computer Science, Florida State University

## ABSTRACT

Text-to-image (T2I) diffusion models have demonstrated impressive image generation capabilities. Still, their computational intensity prohibits resource-constrained organizations from deploying T2I models after fine-tuning them on their internal *target* data. While pruning techniques offer a potential solution to reduce the computational burden of T2I models, static pruning methods use the same pruned model for all input prompts, overlooking the varying capacity requirements of different prompts. Dynamic pruning addresses this issue by utilizing a separate subnetwork for each prompt, but it prevents batch parallelism on GPUs. To overcome these limitations, we introduce Adaptive Prompt-Tailored Pruning (APTP), a novel prompt-based pruning method designed for T2I diffusion models. Central to our approach is a *prompt router* model, which learns to determine the required capacity for an input text prompt and routes it to an architecture code, given a total desired compute budget for prompts. Each architecture code represents a specialized model tailored to the prompts assigned to it, and the number of codes is a hyperparameter. We train the prompt router and architecture codes using contrastive learning, ensuring that similar prompts are mapped to nearby codes. Further, we employ optimal transport to prevent the codes from collapsing into a single one. We demonstrate APTP's effectiveness by pruning Stable Diffusion (SD) V2.1 using CC3M and COCO as *target* datasets. APTP outperforms the single-model pruning baselines in terms of FID, CLIP, and CMMD scores. Our analysis of the clusters learned by APTP reveals they are semantically meaningful. We also show that APTP can automatically discover previously empirically found challenging prompts for SD, *e.g.,* prompts for generating text images, assigning them to higher capacity codes. Our code is available here.

## 1 INTRODUCTION

In recent years, diffusion models (Sohl-Dickstein et al., 2015; Ho et al., 2020; Song & Ermon, 2019; Song et al., 2021b) have revolutionized generative modeling. For instance, modern Text-to-Image (T2I) diffusion models (Adobe FireFly, 2023; Midjourney, 2023; Betker et al., 2023) like Stable Diffusion (Rombach et al., 2022; Podell et al., 2024) have achieved remarkable success in generating realistic images (Nichol et al., 2021; Saharia et al., 2022; Ramesh et al., 2022) and editing them (Kim et al., 2022; Hertz et al., 2022; Zhang et al., 2023b). Consequently, their integration into various applications is of great interest. However, the sampling process of T2I diffusion models is slow and computationally intensive, making them expensive to deploy on GPU clouds for a large number of users and preventing their utilization on edge devices. Thus, reducing the computational cost of T2I models is essential prior to serving them.

Two orthogonal factors underlie the sampling cost of T2I diffusion models: their large number of denoising steps and their parameter-heavy backbone architectures. Most acceleration methods for T2I models reduce the computation per sampling step or skip steps using the techniques such as distillation (Meng et al., 2023; Salimans & Ho, 2022; Habibian et al., 2023) and improved noise

---

*Authors Contributed Equally. Correspondence to {`aliganj, rezashkv`}`@cs.umd.edu`

†Partially supported by NSF IIS 2347592, 2348169, DBI 2405416, CCF 2348306, CNS 2347617.

Figure 1: **Overview:** We prune a text-to-image diffusion model like Stable Diffusion (left) into a mixture of efficient experts (right) in a prompt-based manner. Our prompt router *routes* distinct types of prompts to different experts, allowing experts' architectures to be separately specialized by removing layers or channels.

schedules (Song et al., 2021a; Nichol & Dhariwal, 2021). Other ideas address the second factor and propose efficient architectures for T2I models. Architecture modification methods (Zhao et al., 2023; Kim et al., 2023) modify the U-Net (Ronneberger et al., 2015) of Stable Diffusion (SD) (Rombach et al., 2022) to reduce its parameters and FLOPs while preserving performance. Still, they only provide a few architectural configurations, and generalizing their design choices to new compute budgets or datasets is highly non-trivial. Search-based methods (Li et al., 2024; Liu et al., 2023a) search for an efficient architecture (Li et al., 2024) or an efficient mixture (Liu et al., 2023a) of pretrained T2I models in a model zoo. Yet, evaluating each action in the search process and gathering a model zoo of T2I models are both extremely costly, making search methods impractical in resource-constrained scenarios.

The efficient architecture design methods (Zhao et al., 2023; Kim et al., 2023; Li et al., 2024) have shown promise, but they aim to develop 'one-size-fits-all' architectures for *all* applications. We argue that this approach is misaligned with the common practice for two reasons: 1) Organizations typically fine-tune pretrained T2I models (*e.g.,* SD) on their proprietary *target* data and deploy the resulting model for their specific application. They prioritize performance on their *target* data distribution while meeting their computation budget, and the trade-off between efficiency and performance can vary depending on the complexity of the *target* dataset between organizations. 2) Verifying the validity of each design choice on large-scale datasets used in one-size-fits-all methods is costly and slow to iterate, making them impractical.

Model pruning (Cheng et al., 2023) can reduce a model's computational burden to any desired budget with significantly less effort than designing (Zhao et al., 2023; Kim et al., 2023) or searching (Li et al., 2024) for efficient architectures. Still, T2I models have unique characteristics making existing pruning techniques unsuitable for them. Static pruning methods are input-agnostic and use the same pruned model for all inputs, but distinct prompts of T2I models may require different model capacities. Dynamic pruning employs a separate model for each input sample, but it cannot benefit from batch-parallelism in modern hardware like GPUs and TPUs.

In this paper, we introduce Adaptive Prompt-Tailored Pruning (APTP), a novel prompt-based pruning method for T2I diffusion models. APTP prunes a T2I model pretrained on a large-scale dataset (*e.g.,* SD) using a smaller *target* dataset given a desired compute budget. It tackles the challenges of static and dynamic pruning methods for T2I models by training a *prompt router* module along with a set of *architecture codes*. The prompt router learns to route an input prompt to an architecture code, determining the sub-architecture, called *expert*, of the T2I model to use. Each expert specializes in generating images for the prompts assigned to it by the prompt router (Fig. 1), and the number of experts is a hyperparameter. We train the prompt router and architecture codes using a contrastive learning objective that regularizes the prompt router to select similar architecture codes for similar prompts. In addition, we employ optimal transport to diversify the architecture codes and the resulting experts' budgets, allowing deploying them on hardware with varying capabilities. We take CC3M (Sharma et al., 2018) and MS-COCO (Lin et al., 2014) as the *target* datasets and prune Stable Diffusion V2.1 (Rombach et al., 2022) using APTP in our experiments. APTP outperforms the single-model pruning baselines, and we show that our prompt router learns to group the input prompts into semantic clusters. Further, our analysis demonstrates that APTP can automatically discover challenging prompts for SD, such as prompts for generating text images, found empirically by prior work (Chen et al., 2024; Yang et al., 2024). We summarize our contributions as follows:

- We introduce APTP, a novel prompt-based pruning method for T2I diffusion models. It is more suitable than static pruning for T2I models as it is not input-agnostic. In addition, APTP enables batch parallelism on GPUs, which is not possible with dynamic pruning.

- APTP trains a prompt router and a set of architecture codes. The prompt router maps an input prompt to an architecture code. Each architecture code resembles a pruned sub-architecture expert of the T2I model, specialized in handling certain types of prompts assigned to it.

- We develop a framework to train the prompt router and architecture codes using contrastive learning and employ optimal transport to diversify the architecture codes and their corresponding sub-architectures' computational requirements, given a desired compute budget.

## 2 RELATED WORK

Several works have addressed improving the architectural efficiency of diffusion models, which is our paper's primary focus. **Multi-expert** (Ganjdanesh et al., 2024c; Zhang et al., 2023a; Liu et al., 2023a; Pan et al., 2024; Xue et al., 2023) methods employ several models each responsible for an interval of diffusion model's denoising process. These methods design expert model architectures (Lee et al., 2024; Zhang et al., 2023a; Xue et al., 2023), train several models with varying capacities from scratch (Liu et al., 2023a), or utilize existing pretrained models (Liu et al., 2023a; Pan et al., 2024). However, pretrained experts may not be available, and training several models from scratch, as done by multi-expert methods, is prohibitively expensive. **Architecture Design** (Zhao et al., 2023; Yang et al., 2023a; Kim et al., 2023) approaches redesign the U-Net architecture (Ronneberger et al., 2015) of diffusion models to enhance its efficiency. MobileDiffusion (Zhao et al., 2023) does so using empirical heuristics derived from performance metrics on MS-COCO (Lin et al., 2014). BK-SDM (Kim et al., 2023) removes some blocks from the SD's U-Net and applies distillation (Hinton et al., 2015) to train the pruned model. Spectral Diffusion (Yang et al., 2023a) introduces a wavelet gating operation and performs frequency domain distillation. Yet, generalizing heuristics and design choices of the architecture design methods to other tasks and compute budgets is non-trivial. Alternatively, SnapFusion (Li et al., 2024) searches for an efficient architecture for T2I models. Yet, evaluating each action in the search process requires 2.5 A100 GPU hours, making it costly in practice. Finally, SPDM (Fang et al., 2023) estimates the importance of different weights using Taylor expansion and removes the low-scored ones. Despite their promising results, all these methods are 'static' in that they obtain an efficient model and utilize it for *all* inputs. This is suboptimal for T2I models as input prompts may vary in complexity, demanding different model capacity levels. Our method differs from existing approaches by introducing a prompt-based pruning technique for T2I models. It is the first method that allocates computational resources to prompts based on their individual complexities while ensuring optimal utilization and batch-parallelizability. See Appendix B for a detailed review of related work, with more focus on sampling efficiency studies that are orthogonal to our approach.

## 3 METHOD

We introduce a framework for prompt-based pruning of T2I diffusion models termed Adaptive Prompt-Tailored Pruning (ATPT). APTP prunes a T2I model pretrained on large-scale datasets (*e.g.*, Stable Diffusion (Rombach et al., 2022)) using a smaller *target* dataset. This approach mirrors the common practice where organizations fine-tune pretrained T2I models on their internal proprietary data and deploy the resulting model for their customers. The core component of APTP is a *prompt router* that learns to map an input prompt to an *architecture code* during the pruning process. Each architecture code corresponds to a specialized *expert* model, which is a sub-network of the T2I model. We train the prompt router and architecture codes in an end-to-end manner. After the pruning phase, APTP fine-tunes each specialized model using samples from the target dataset assigned to it by the prompt router. We elaborate on the components of APTP in the following subsections.

### 3.1 BACKGROUND

Given an input text prompt, text-to-image (T2I) diffusion models generate a corresponding image by iteratively denoising a Gaussian noise (Ho et al., 2020; Song & Ermon, 2019; Sohl-Dickstein et al., 2015). They achieve this by training a denoising model, $\epsilon(\cdot; \theta)$, parameterized by $\theta$. For a given training image-text pair $(x_0, p) \sim \mathcal{P}$, T2I models define a forward diffusion process, progressively adding Gaussian noise to the initial image $x_0$ over $T$ steps. This process is defined as $q(x_t|x_0) = \mathcal{N}(x_t; \sqrt{\bar{\alpha}_t}x_0, (1 - \bar{\alpha}_t)I)$, where $\bar{\alpha}_t$ is the forward noise schedule parameter, typically chosen such that $q(x_t|x_0) \rightarrow \mathcal{N}(0, I)$ as $t \rightarrow T$. T2I models are trained using the variational evidence lower bound (ELBO) objective (Ho et al., 2020):

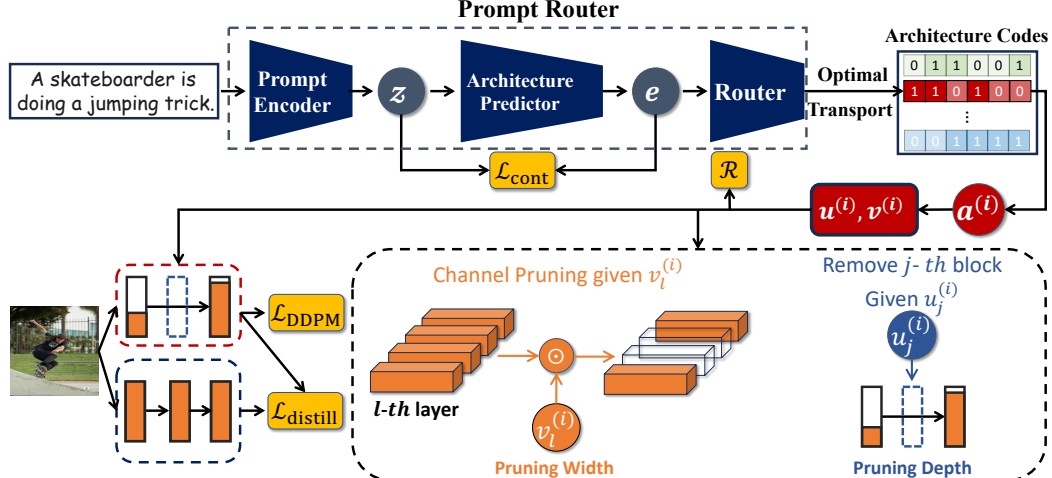

Figure 2: **Our pruning scheme.** We train our prompt router and the set of architecture codes to prune a text-to-image diffusion model into a mixture of experts. The prompt router consists of three modules. We use a Sentence Transformer (Reimers & Gurevych, 2019) as our prompt encoder to encode the input prompt into a representation $z$. Then, the architecture predictor transforms $z$ into the architecture embedding $e$ that has the same dimensionality as architecture codes. Finally, the router routes the embedding $e$ into an architecture code $a^{(i)}$. We use optimal transport to evenly assign the prompts in a training batch to the architecture codes. The architecture code $a^{(i)} = (u^{(i)}, v^{(i)})$ determines pruning the model's width and depth. We train the prompt router's parameters and architecture codes in an end-to-end manner using the denoising objective of the pruned model $\mathcal{L}_{\text{DDPM}}$, distillation loss between the pruned and original models $\mathcal{L}_{\text{distill}}$, average resource usage for the samples in the batch $\mathcal{R}$, and contrastive objective $\mathcal{L}_{\text{cont}}$, encouraging embeddings $e$ preserving semantic similarity of the representations $z$.

$$\mathcal{L}_{\text{DDPM}}(\theta) = \mathbb{E}_{\substack{(x_0,p)\sim\mathcal{P} \\ t\sim[1,T] \\ \epsilon\sim\mathcal{N}(0,I) \\ x_t\sim q(x_t|x_0)}} ||\epsilon(x_t, p, t; \theta) - \epsilon||^2 \qquad (1)$$

T2I models sample an image starting from a Gaussian noise $x_T \sim \mathcal{N}(0, I)$ and denoising it using the trained denoising model. We refer to appendix for a thorough review of diffusion models.

## 3.2 PROMPT ROUTER AND ARCHITECTURE CODES

Our approach prunes a diffusion model into a *Mixture of Experts*, where each expert specializes in handling a distinct group of prompts with different complexities. Each expert corresponds to a unique sub-network $a^{(i)} \in \{0, 1\}^D$, where $D$ represents the total number of prunable units in the T2I model. These sub-networks are optimized to efficiently process their assigned prompt groups while maintaining high performance. To assign prompts to the appropriate expert sub-network, we design a *prompt router model* comprising three key components:

1. **Prompt Encoder**: This module encodes input prompts into semantically meaningful embeddings. Prompts with similar semantics are mapped to embeddings that are close in the embedding space, ensuring that similar prompts are routed to similar sub-networks.

2. **Architecture Predictor**: The encoded prompt embeddings are further transformed into architecture embeddings $e$. This step bridges the gap between the high-level semantics of prompts and the architectural configurations needed to process them efficiently.

3. **Router Module**: The architecture embeddings $e$ are finally mapped to specific architecture codes $a$, which define the structure of the sub-network (expert) that will handle the prompt.

The remainder of this section provides detailed descriptions of how each component functions and contributes to the overall pruning and routing process.

We employ our prompt router to determine the specialized sub-network of the T2I model to be used for a given input prompt. Specifically, we denote our prompt router with the function $f_{\text{PR}}(\cdot; \eta, \mathcal{A})$, parameterized by $\eta$, which maps an input prompt $p$ to an architecture code $a$:

$$a = f_{\text{PR}}(p; \eta, \mathcal{A}) \tag{2}$$

We define the set of learnable architecture codes as $\mathcal{A} = \{a^{(i)}\}_{i=1}^{N}$, where $N$ is a hyperparameter. Also, $a^{(i)} \in \mathbb{R}^D$, with $D$ being the number of prunable width and depth units in the T2I model. Given a desired constraint $T_d$ on the total compute budget (latency, MACs, etc.) for the prompts in the target dataset, we train the prompt router's parameters $\eta$ and architecture codes in $\mathcal{A}$ to obtain a set of expert models. These experts are specialized, efficient, and performant sub-networks of the T2I model, as determined by the architecture codes. As illustrated in Fig. 2, the prompt router consists of a prompt encoder, an architecture predictor, and a router module. We provide details of these components in the following subsections and present our pruning objective in Eq. 15.

### 3.2.1 PROMPT ENCODER AND ARCHITECTURE PREDICTOR

Our primary intuition in designing the prompt router is that it should route semantically similar prompts to similar sub-networks of a T2I model. Accordingly, we use a pretrained frozen Sentence Transformer model (Reimers & Gurevych, 2019) as our prompt encoder module. It can effectively encode input prompts $p$ into semantically meaningful embeddings $z$:

$$z = f_{\text{PE}}(p) \tag{3}$$

$f_{\text{PE}}(\cdot)$ is the prompt encoder. We do not explicitly show the prompt encoder's parameters as we do not train them in our pruning method. The Sentence Transformer can encode semantically similar prompts to nearby embeddings and distant from dissimilar ones. We leverage this property in our framework (Eq. 13) to ensure the prompt router maps similar prompts to similar architecture codes.

The architecture predictor module transforms prompt embeddings $z$ into architecture embeddings $e$:

$$e = f_{\text{AP}}(z; \eta) \tag{4}$$

$\eta$ denotes the parameters of the architecture predictor $f_{\text{AP}}(\cdot)$, implemented with a single feed-forward layer (more details in Appendix D). The embeddings $e$ have the same dimensionality as the architecture codes $a$.

### 3.2.2 ROUTER

The router module takes an architecture embedding $e$ and routes it to an architecture code $a \in \mathcal{A}$. A straightforward way to implement the router is to map an input embedding $e$ to its nearest architecture code in $\mathcal{A}$. However, we found that this approach may lead to the collapse of architecture codes, as they could converge to a single code that meets the desired compute budget $T_d$ with a relatively decent performance, causing the prompt router to route all input prompts to it.

To tackle this challenge, we employ optimal transport in our router module during the pruning phase. Formally, let $E = [e_1, \cdots, e_B]$ represent $B$ architecture embeddings in a training batch, and $A = [a^{(1)}, \cdots, a^{(N)}]$ represent the $N$ codes. The goal is to find an assignment matrix $Q = [q_1, \cdots, q_B]$ that maximizes the similarity between architecture embeddings and their assigned architecture codes:

$$\max_{Q \in \mathcal{Q}} \text{Tr}(Q^T A^T E) + \epsilon H(Q) \tag{5}$$

H is an entropy term $H(Q) = -\sum_{i,j} q_{ij} \log(q_{ij})$ and $\epsilon$ is the regularization strength. It has been shown (YM. et al., 2020; Caron et al., 2020) that high values of $\epsilon$ lead to a uniform assignment matrix $Q$, causing all codes in $A$ to collapse to a single code. Thus, we set $\epsilon$ to a small value. Further, we impose an equipartition (YM. et al., 2020) constraint on $Q$ so that architecture embeddings in a batch are assigned equally to architecture codes:

$$\mathcal{Q} = \{Q \in \mathbb{R}_+^{N \times B} \mid Q 1_B = \frac{1}{N} 1_N, \ Q^T 1_N = \frac{1}{B} 1_B\} \tag{6}$$

Here, $1_{\{B,N\}}$ are vectors of ones with length $B$ and $N$, respectively. These constraints enforce that, on average, $\frac{B}{N}$ embeddings are assigned to an architecture code per training batch, thereby ensuring that each architecture code gets enough samples for training. The optimal transport problem in Eq. 5 with the constraints in Eq. 6 can be solved using the fast version (Cuturi, 2013) of the Sinkhorn-Knopp algorithm and the solution has the normalized exponential matrix form (Cuturi, 2013):

$$Q^* = \text{diag}(m)\exp(\frac{A^T E}{\epsilon})\text{diag}(n) \tag{7}$$

Here, $m$ and $n$ are renormalization vectors that can be calculated using a few iterations of the Sinkhorn-Knopp algorithm. With $\epsilon$ set to a small value, the matrix $BQ^*$ will have columns that are close to one-hot vectors, which we use to assign the architecture embeddings to the architecture codes. In summary, the router module's function during the pruning process is:

$$a = f_{\text{R}}(e, \mathcal{A}; Q^*) \tag{8}$$

After the pruning stage, the trained router simply routes an input architecture embedding to an architecture code with the highest cosine similarity. We provide the definition of $f_{\text{R}}$ in Appendix D.2.

### 3.3 PRUNING

We divide each architecture code $a^{(i)} \in \mathcal{A}$ into two sub-vectors $a^{(i)} = (u^{(i)}, v^{(i)})$. We utilize the vectors $u^{(i)}$ and $v^{(i)}$ to prune the depth layers and determine widths of the layers, respectively.

A simple approach to prune the width of a layer (a depth layer) is to use binary vectors $v^{(i)}$ ($u^{(i)}$), indicating whether the channels (layers) should be pruned. Yet, doing so is not differentiable, and one needs to solve a discrete optimization problem to find the optimal binary vectors. Instead, we employ soft vectors $\mathbf{v}^{(i)}$ ($\mathbf{u}^{(i)}$) that are continuous and differentiable for pruning. We calculate them as:

$$\mathbf{v}^{(i)} = \text{sigmoid}(\frac{v^{(i)} + g_v}{\gamma}), \quad \mathbf{u}^{(i)} = \text{sigmoid}(\frac{u^{(i)} + g_u}{\gamma}) \tag{9}$$

$g_{\{u,v\}} \sim \text{Gumbel}(0,1)$ represents a noise vector sampled from the Gumbel distribution (Gumbel, 1954), and $\gamma$ is the temperature. This formulation, known as the Gumbel-sigmoid reparameterization (Jang et al., 2017; Maddison et al., 2017), is a differentiable approximation of sampling from Bernoulli distributions with parameters `sigmoid`$(v^{(i)})$ and `sigmoid`$(u^{(i)})$. When the temperature $\gamma$ is set appropriately, the vectors $\mathbf{v}^{(i)}$ and $\mathbf{u}^{(i)}$ will be close to binary vectors, and we use them for pruning the layers' width and depth layers. Assuming $\mathbf{v}^{(i)} = [\mathbf{v}_l^{(i)}]_{l=1}^L$ where $L$ is the number of the model's layers, we prune the width of the $l$-th layer as:

$$\widehat{\mathcal{F}}_l = \mathcal{F}_l \odot \mathbf{v}_l^{(i)} \tag{10}$$

In this equation, $\mathcal{F}_l$ represents feature maps of the $l$-th layer, and $\odot$ is the element-wise multiplication. We prune the channels of the convolution layers in the ResBlocks (He et al., 2016), attention heads in the Transformer layers (Vaswani et al., 2017), and the channels of the feed-forward layers in the Transformer layers.

In a similar manner, we apply the vectors $\mathbf{u}^{(i)} = [\mathbf{u}_j^{(i)}]_{j=1}^M$ (where M is the number of depth layers that we prune) for pruning the model's depth. Specifically, we prune the $j$-th depth layer $f_j$ as:

$$\widehat{\mathcal{F}}_j = \mathbf{u}_j^{(i)} f_j(\mathcal{F}_{j-1}) + (1 - \mathbf{u}_j^{(i)})\mathcal{F}_{j-1} \tag{11}$$

$\mathcal{F}_{j-1}$ denotes the previous layer's feature maps. The granularity of our depth pruning is a ResBlock or a Transformer layer in the U-Net of the T2I model. We provide more details in Appendix D.3. In summary, we employ the architecture vectors $\mathbf{a}^{(i)} = (\mathbf{v}^{(i)}, \mathbf{u}^{(i)})$ to prune the T2I model.

### 3.3.1 TRAINING THE PROMPT ROUTER AND ARCHITECTURE CODES

We jointly train the prompt router and architecture codes in an end-to-end manner, guiding the prompt router to map similar prompts to similar architecture codes. In addition, we regularize the architecture codes to correspond to performant sub-networks of the T2I model, be diverse, and adhere to the desired compute budget $T_d$ on aggregate.

**Contrastive Training.** Given a training batch with $B$ prompts, we compute their prompt embeddings $z$ (Eq. 3) and architecture embeddings $e$ (Eq. 4). We calculate architecture vectors $\mathbf{e}'$:

$$\mathbf{e}' = \text{sigmoid}(\frac{e+g}{\gamma}) \tag{12}$$

where $\gamma$ and $g$ have the same definitions as Eq. 9. We define the cosine similarity of two vectors as $\text{sim}(\mathbf{m}, \mathbf{n}) = \mathbf{m}^T\mathbf{n}/||\mathbf{m}|| \cdot ||\mathbf{n}||$, and we use the following objective to train the architecture predictor:

$$\mathcal{L}_{\text{cont}}(\eta) = \frac{1}{B^2} \sum_{i=1}^{B} \sum_{j=1}^{B} r_{i,j}\log(s_{i,j}) + (1 - r_{i,j})\log(1 - s_{i,j}) \tag{13}$$

$$r_{i,j} = \frac{\exp(\text{sim}(z_i, z_j)/\tau)}{\sum_{k=1}^{B} \exp(\text{sim}(z_i, z_k)/\tau)}, \quad s_{i,j} = \frac{\exp(\text{sim}(\mathbf{e}'_i, \mathbf{e}'_j)/\tau)}{\sum_{k=1}^{B} \exp(\text{sim}(\mathbf{e}'_i, \mathbf{e}'_k)/\tau)} \tag{14}$$

$\tau$ is a temperature parameter. Eq. 13 regularizes the architecture predictor to map representations $z$ to the regions of the space of the architecture embeddings $e$ such that their corresponding architecture vectors $\mathbf{e}'$ maintain the similarity between the prompts. Note that we do not actually use the architecture vectors $\mathbf{e}'$ to prune the model (Sec. 3.2.2, 3.3). Yet, we apply our contrastive regularization to the vectors $\mathbf{e}'$ instead of embeddings $e$. This design choice is a result of our observation that applying $\mathcal{L}_{\text{cont}}$ to embeddings $e$ may not lead to diverse architecture vectors $\mathbf{a}^{(i)} = (\mathbf{v}^{(i)}, \mathbf{u}^{(i)})$ that we use for pruning (Sec. 3.3). For instance, embeddings $e$ (and their nearby architecture codes $a$) can be distributed in the embedding space (before Gumbel-Sigmoid estimation), but all be in saturation regions of the sigmoid function (Eqs. 9, 12), resulting in similar architecture vectors $\mathbf{e}'$ and $\mathbf{a}^{(i)}$.

In contrast, Eq. 13 implicitly diversifies the architecture vectors $\mathbf{a}^{(i)}$. The reason is that the router routes embeddings $e$ to codes $a$ (Eq. 8) that have high similarity with each other (Eq. 7). Also, Eq. 13 distributes embeddings $e$ in the space of the architecture embeddings such that the architecture vectors $\mathbf{e}'$ become similar (different) for similar (dissimilar) prompts. As the vectors $\mathbf{e}'$ and $\mathbf{a}^{(i)}$ are calculated in a similar manner (Eqs. 9, 12), the diversity of vectors $\mathbf{e}'$ implies the same for vectors $\mathbf{a}^{(i)}$.

Assuming $B^{(i)}$ samples get routed to the architecture code $a^{(i)}$ in a training batch ($\sum_i B^{(i)} = B$), we train the prompt router and the architecture codes using the following objective:

$$\min_{\eta, \mathcal{A}} \mathcal{L} = [\frac{1}{N} \sum_{i=1}^{N} [\frac{1}{B^{(i)}} \sum_{j=1}^{B^{(i)}} [\mathcal{L}_{\text{DDPM}}(x_j^{(i)}, p_j^{(i)}; a^{(i)}) + \lambda_{\text{distill}}\mathcal{L}_{\text{distill}}(x_j^{(i)}, p_j^{(i)}; a^{(i)})]]]$$
$$+ \lambda_{\text{res}}\mathcal{R}(\widehat{T}(\mathcal{A}), T_d) + \lambda_{\text{cont}}\mathcal{L}_{\text{cont}}(\eta) \tag{15}$$

$\mathcal{L}_{\text{DDPM}}((x_j^{(i)}, p_j^{(i)}); a^{(i)})$ denotes the denoising objective for the sample $(x_j^{(i)}, p_j^{(i)})$ routed to subnetwork chosen by the architecture code $a^{(i)}$. $\mathcal{R}(\widehat{T}(\mathcal{A}), T_d)$ regularizes the weighted average of the MACs used by architecture codes ($\widehat{T}(\mathcal{A}) = \sum_i \frac{B^{(i)}}{B}[\widehat{T}(a^{(i)})]$) to be close to $T_d$. We define $\mathcal{R}(x, y) = \log(\max(x, y)/\min(x, y))$, and $\{\lambda_{\text{distill}}, \lambda_{\text{res}}, \lambda_{\text{cont}}\}$ are hyperparameters. Finally, $\mathcal{L}_{\text{distill}}$ is the distillation objective (Kim et al., 2023) regularizing the pruned model having similar outputs to the original one. We refer to Appendix D.4 for more details about our distillation objective.

### 3.4 FINE-TUNING THE PRUNED EXPERT MODELS

After the pruning stage, we use the learned architecture codes to prune the T2I model into our experts. Then, we fine-tune experts using samples routed to them. We use the same training objective as the pretraining stage while adding distillation to fine-tune experts and refer to Appendix D.5 for more details. At test time, our trained prompt router routes an input prompt to one of the experts, and we use the expert to generate an image for it.

## 4 EXPERIMENTS

We use Conceptual Captions 3M (CC3M) (Sharma et al., 2018) and MS-COCO (Lin et al., 2014) as our *target* datasets to prune the Stable Diffusion (SD) V2.1 model to demonstrate APTP's effectiveness. On CC3M, we prune the model with Base: (0.85 MACs, 16 experts) and Small: (0.66

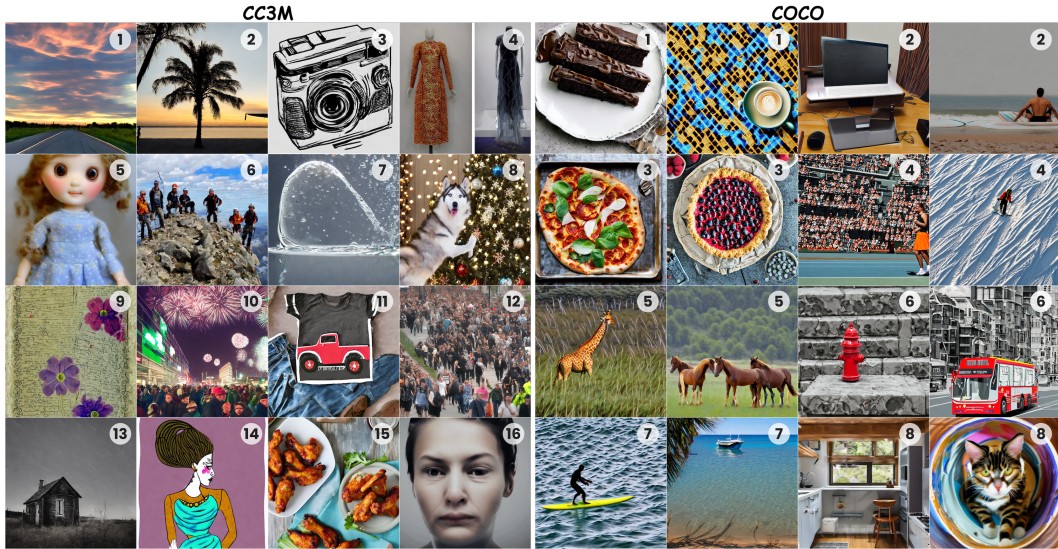

Figure 3: Samples of the APTP-Base experts after pruning the Stable Diffusion V2.1 using CC3M (Sharma et al., 2018) and COCO (Lin et al., 2014) as the *target* datasets. Expert IDs are shown on the top right of images. (See Table 8 for prompts)

MACs, 8 experts) configurations. On MS-COCO, we perform Base: (0.78 MACs, 8 experts) and Small: (0.64 MACs, 8 experts) pruning settings. We set $(\lambda_{\text{distill}}, \lambda_{\text{res}}, \lambda_{\text{cont}})=(0.2, 2.0, 100)$ (Eq. 15) and the temperature $\tau$ (Eq. 14) to 0.03. We evaluate all models with FID (Heusel et al., 2017), CLIP (Hessel et al., 2021), and CMMD (Jayasumana et al., 2023) scores using 14k samples in the validation set of CC3M and 30k samples from the MS-COCO's validation split. For all models, we sample the images at the resolution of 768 and resize them to 256 for calculating the metrics, using the 25-steps PNDM (Liu et al., 2022) sampler following BK-SDM (Kim et al., 2023). We refer to Appendix E for more details.

## 4.1 COMPARISON RESULTS

As APTP is the first pruning method specifically designed to prune a pretrained T2I model on a *target* dataset, we compare its performance with SD V2.1, weight norm pruning (Li et al., 2017), and two recently proposed static pruning baselines, namely Structural Pruning (SP) (Fang et al., 2023) and BKSDM (Kim et al., 2023). Table 1 shows the results. We fine-tune APTP, SP and BKSDM for 30k iterations after pruning and give Norm-pruning 50k fine-tuning iterations to ensure they all reach their final performance level.

**CC3M:** Table 1a summarizes the results on CC3M. With a similar MACs budget and latency values, APTP (0.85) significantly outperforms the Norm pruning (Li et al., 2017), SP (Fang et al., 2023), and BKSDM (Kim et al., 2023) baselines with a significant margin in terms of FID, CLIP, and CMMD scores. It also achieves 15% less latency while showing close performance scores to SD V2.1. Notably, APTP (0.66) has approximately 23% less latency and 21% lower MACs budget than the baselines, but it still outperforms them on all metrics. These results illustrate the advantages of prompt-based compared to static pruning.

**MS-COCO:** We present the results for MS-COCO in Table 1b. APTP (0.78) reduces the latency of SD by 22.5% while preserving its CLIP score and achieving a close CMMD score. Further, with a similar latency, APTP (0.78) significantly outperforms the static pruning baselines with at least 3.71 FID (BKSDM), 2.34 CLIP (SP), and 0.042 CMMD (BKSDM) scores. Similar to the results on CC3M, APTP (0.64) achieves approximately 19.3% less latency than the Norm pruning and SP while outperforming them on all scores. In summary, our quantitative evaluations show the clear advantage of prompt-based compared to static pruning for T2I models. We provide samples of APTP on the validation sets of CC3M and MS-COCO in Fig. 3 and Appendix E.4.6.

Table 1: Results on CC3M and MS-COCO. We report performance metrics using samples generated at the resolution of 768 then downsampled to 256 (Kim et al., 2023). We measure models' MACs/Latency with the input resolution of 768 on an A100 GPU. @30/50k shows fine-tuning iterations after pruning.

| Method | CC3M Complexity | | Performance | | | Method | MS-COCO Complexity | | Performance | | |
| --- | --- | --- | --- | --- | --- | --- | --- | --- | --- | --- | --- |
| | MACs (@768) | Latency (↓) (Sec/Sample) (@768) | FID (↓) | CLIP (↑) | CMMD (↓) | | MACs (@768) | Latency (↓) (Sec/Sample) (@768) | FID (↓) | CLIP (↑) | CMMD (↓) |
| Norm (Li et al., 2017) @50k | 1185.3G | 3.4 | 141.04 | 26.51 | 1.646 | Norm (Li et al., 2017) @50k | 1077.4G | 3.1 | 47.35 | 28.51 | 1.136 |
| SP (Fang et al., 2023) @30k | 1192.1G | 3.5 | 75.81 | 26.83 | 1.243 | SP (Fang et al., 2023) @30k | 1071.4G | 3.3 | 53.09 | 28.98 | 0.926 |
| BKSDM (Kim et al., 2023) @30k | 1180.0G | 3.3 | 87.27 | 26.56 | 1.679 | BKSDM (Kim et al., 2023) @30k | 1085.4G | 3.1 | 26.31 | 28.89 | 0.611 |
| APTP(0.66) @30k | 916.3G | 2.6 | 60.04 | 28.64 | 1.094 | APTP(0.64) @30k | 890.0G | 2.5 | 39.12 | 29.98 | 0.867 |
| APTP(0.85) @30k | 1182.8G | 3.4 | 36.77 | 30.84 | 0.675 | APTP(0.78) @30k | 1076.6G | 3.1 | 22.60 | 31.32 | 0.569 |
| SD 2.1 | 1384.2G | 4.0 | 32.08 | 31.12 | 0.567 | SD 2.1 | 1384.2G | 4.0 | 15.47 | 31.33 | 0.500 |
| (a) | | | | | | (b) | | | | | |

Table 2: The most frequent words in prompts assigned to each expert of APTP-Base pruned on CC3M. The resource utilization of each expert is indicated in parentheses.

| Expert 1 (0.72) | Expert 2 (0.73) | Expert 3 (0.75) | Expert 4 (0.76) |
| --- | --- | --- | --- |
| View - Sunset - City - Building - Sky | View - Boat - Sea | Artist - Actor | Actor - Dress - Portrait |
| **Expert 5 (0.77)** | **Expert 6 (0.78)** | **Expert 7 (0.79)** | **Expert 8 (0.79)** |
| Illustration - Portrait - Photo | Player - Ball - Game - Team | Background - Water - River - Tree | Biological Species - Dog - Cat |
| **Expert 9 (0.79)** | **Expert 10 (0.80)** | **Expert 11 (0.81)** | **Expert 12 (0.81)** |
| Illustration - Vector | People | Car - City - Road | Person - Player - Team - Couple |
| **Expert 13 (0.86)** | **Expert 14 (0.90)** | **Expert 15 (0.95)** | **Expert 16 (0.98)** |
| Room - House | Art - Artist - Digital | Food - Water | Person - Man - Woman - Text |

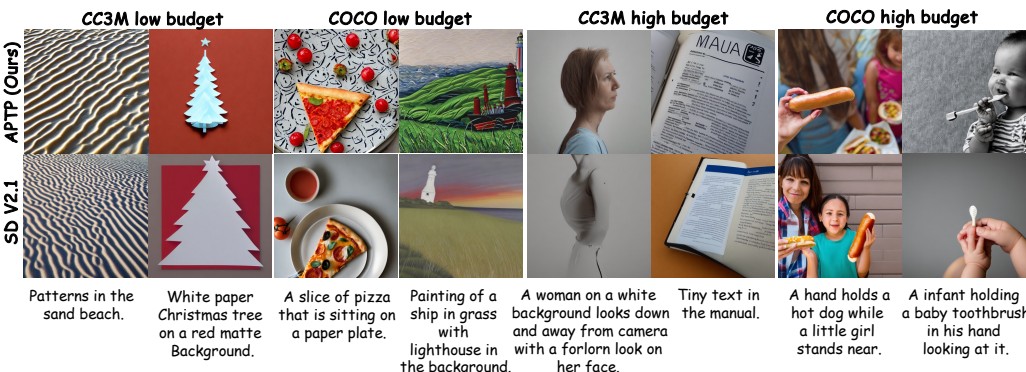

Figure 4: Comparison of samples generated by low and high budget experts of APTP-Base *vs.* SD V2.1 on CC3M and MS-COCO validation sets.

## 4.2 ANALYSIS OF THE PROMPT ROUTER

We analyze our prompt router's behavior by examining the prompts it assigns to experts in the APTP-Base (0.85) model for the CC3M experiment. Table 2 displays the most frequent words in the prompts routed to each expert, along with the experts' MACs budgets. Our prompt router effectively "specializes" experts by assigning distinct topics to experts with varying budgets. For instance, Expert 1 focuses on cityscapes, Expert 8 on animals, and Expert 13 on house interiors and exteriors. Notably, the prompt router assigns images of textual content and human beings to Expert 16, which has the highest budget. These categories have been empirically found to be challenging for SD 2.1 (Chen et al., 2024; Yang et al., 2024), and our prompt router can automatically discover and route them to higher-capacity experts. In contrast, paintings and illustrations are assigned to lower-budget experts, as they seem to be easier for the model to generate. We provide examples of high and low resource prompts for APTP-Base on both CC3M and COCO in Fig. 4.

## 4.3 ABLATION STUDY

We conduct two ablation experiments to study the impact of APTP's components on its performance. First, we implement a naive baseline that uses parameterizations $\mathbf{v} = \text{sigmoid}((\eta_v + g_v)/\tau)$ and $\mathbf{u} = \text{sigmoid}((\eta_u + g_u)/\tau)$ (Eq. 9) to prune a single model. It directly trains two vectors $(\eta_v, \eta_u)$ to do so ('Uni-Arch Baseline' in Table 3). Then, we start with a router trained only using the contrastive objective (Eq. 13) and add optimal transport and distillation incrementally to prune SD V2.1 into 8 experts with 80% MACs budget on MS-COCO (Lin et al., 2014). We fine-tune all pruned models with 10k iterations, and Table 3 presents the results. We observe that the contrastive training (Eq. 13) alone fails to improve results from pruning a single model to a mixture of experts. This is because although the contrastive objective makes the architecture codes diverse, it does not enforce the prompt router to distribute the prompts between architecture codes. Thus, it routes most of the input prompts to a single expert. As a result, all experts except one receive insufficient training samples and generate low-quality samples after fine-tuning, leading to the mixture showing worse metrics than the baseline. Employing optimal transport (Eq. 5) in the prompt router significantly improves FID (10.22), CLIP (1.17), and CMMD (0.18) scores of the mixture. In addition, distillation can further improve the architecture search process of the experts, resulting in a more performant mixture. In summary, these results validate the effectiveness of our design choices for APTP.

Table 3: Ablation results of APTP's components on 30k samples from MS-COCO (Lin et al., 2014) validation set. We fine-tune all models for 10k iterations after pruning.

| Method | MACs(@768) | Latency(@768) | FID ($\downarrow$) | Clip Score ($\uparrow$) | CMMD ($\downarrow$) |
|---|---|---|---|---|---|
| Uni-Arch Baseline | 1088.8G | 3.1 | 46.56 | 29.11 | 0.91 |
| Contrastive Router | 1079.5G | 3.1 | 48.78 | 28.90 | 0.92 |
| + Optimal Transport | 1076.6G | 3.1 | 38.56 | 30.07 | 0.74 |
| + Distillation (**APTP**) | 1076.6G | 3.1 | 25.57 | 31.13 | 0.58 |

In our second ablation experiment, we explore the impact of the number of experts on APTP. We prune SD V2.1 to 80% MACs budget using APTP with 4, 8, and 12 experts on MS-COCO. Fig. 5 shows the results. Interestingly, the results demonstrate that the relationship between FID and CLIP scores with the number of experts is nonlinear, and the optimal number of experts is dataset-dependent. This observation illustrates that prompt-based pruning is more suitable than static pruning for T2I models.

## 5 CONCLUSION

In this paper, we develop Adaptive Prompt-Tailored Pruning (APTP), the first prompt-based pruning method for text-to-image (T2I) diffusion models. APTP takes a T2I model, pretrained on large-scale data, and prunes it using a *target* dataset, resembling the common practice that organizations fine-tune T2I models on their internal data before deployment. The core element of APTP is a prompt router module that learns to decide the model capacity required to generate a sample for an input prompt, routing it to an architecture code given a desired compute budget. Each architecture code corresponds to a sub-network of the T2I model, specializing in generating images for the prompts that prompt router routes to it. APTP trains the prompt router and architecture codes in an end-to-end manner, encouraging the prompt router to route similar prompts to similar architecture codes. Further, we utilize optimal

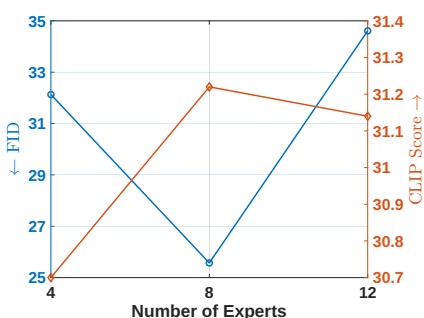

Figure 5: Ablation Results for the number of experts of APTP on MS-COCO.

transport in the prompt router of APTP during pruning to diversify the architecture codes. Our experiments in which we prune Stable Diffusion (SD) V2.1 using CC3M and MS-COCO as *target* datasets demonstrate the benefit of prompt-based pruning compared to conventional static pruning methods for T2I models. Further, our analysis on APTP's prompt router reveals that it can automatically discover challenging prompt types for SD, like generating text, humans, and fingers, routing them to experts with high compute budgets.

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

## A   LIMITATIONS AND BROADER IMPACTS

We believe a limitation of our work may be that our experimental scope is relatively limited. We chose CC3M Sharma et al. (2018) and MS-COCO Lin et al. (2014) as they contain at least 100k image-text pairs and are diverse to enable us to simulate the real-world case of fine-tuning a pretrained T2I model like SD v2.1 on a *target* dataset before deploying it. Yet, exploring APTP's performance on other tasks is an interesting future work. In addition, APTP, like Stable Diffusion, has challenges generating images of text, humans, and fingers (Sec. 4.2, Fig. 4), and one can employ techniques in recent works Chen et al. (2024); Yang et al. (2024); Gandikota et al. (2023) to alleviate these issues. Finally, we note that our method can enable organizations to serve their models with a lower cost for GenAI applications, but it may also facilitate harmful content generation, necessitating responsible usage of generative models.

## B   RELATED WORK

**Efficient Diffusion Models:** Methods for accelerating and improving the efficiency of diffusion models fall into two categories. The first group focuses on reducing the complexity or the number of sampling steps of diffusion models. They use techniques like distillation Habibian et al. (2023); Meng et al. (2023); Salimans & Ho (2022); Song et al. (2024); Liu et al. (2024a), learning optimal denoising time-steps Watson et al. (2022b;a), designing improved noise schedules Nichol & Dhariwal (2021); Song et al. (2021a); Zhang et al. (2023c), caching intermediate computations Ma et al. (2023); Agarwal et al. (2023); Wimbauer et al. (2023), and proposing faster solvers Lu et al. (2023); Xu et al. (2023); Liu et al. (2023b). The second group is aimed at improving the architectural efficiency of diffusion models, which is the main focus of our paper. **Multi-expert** Lee et al. (2024); Zhang et al. (2023a); Liu et al. (2023a); Pan et al. (2024); Xue et al. (2023) methods employ multiple expert models each responsible for a separate part of the denoising process of diffusion models. These methods either design expert model architectures Lee et al. (2024); Zhang et al. (2023a); Xue et al. (2023), train several models with varying capacities from scratch Liu et al. (2023a), or utilize existing pretained experts Liu et al. (2023a); Pan et al. (2024) with different capacities. However, pretrained experts are not necessarily available, and training several models from scratch, as required by multi-expert methods, is prohibitively expensive in practice. **Architecture Design** approaches Zhao et al. (2023); Yang et al. (2023a); Kim et al. (2023) redesign the architecture of diffusion models to enhance efficiency. MobileDiffusion Zhao et al. (2023) modifies the U-Net Ronneberger et al. (2015) architecture of Stable Diffusion (SD) based on empirical heuristics derived from the model's performance on the MS-COCO Lin et al. (2014) dataset. BK-SDM Kim et al. (2023) removes some blocks from the U-Net model of SD and applies knowledge distillation from the original SD model to the pruned model. Spectral Diffusion Yang et al. (2023a) introduces a wavelet gating operation and performs frequency domain distillation from a pretrained teacher model into a small student model. Yet, generalization of the heuristics and design choices in the architecture design methods to other tasks and compute budgets is non-trivial. **Quantization** methods So et al. (2024); He et al. (2024); Pandey et al. (2023); Yang et al. (2023b); Liu et al. (2024b); Tang et al. (2023); Chu et al. (2024); Li et al. (2023) reduce the precision of model weights and activations during the forward pass to accelerate the sampling process. Different from these methods, SnapFusion Li et al. (2024) searches for an efficient architecture for T2I models. However, evaluating each action in the search process consumes about 2.5 A100 GPU hours, rendering it impractical for resource-constrained scenarios. Finally, SPDM Fang et al. (2023) estimates the importance of different weights in the model using Taylor expansion and removes the low-scored architectures. Despite their promising results, all these methods are 'static' in that they obtain an efficient model and utilize it for all inputs. This is suboptimal for T2I models as input prompts may vary in complexity, demanding different model capacity levels.

**Pruning and Neural Architecture Search (NAS):** Our paper also intersects with model pruning Cheng et al. (2023); Ganjdanesh et al. (2022; 2024b) and NAS Zoph & Le (2017); Liu et al. (2019); Cai et al. (2020); Yao et al. (2021); Hou et al. (2020); White et al. (2023); Ganjdanesh et al. (2023) methods. These methods prune pretrained models and search for suitable architectures given a specific task and computational budget. Existing pruning techniques can be categorized into Static and Dynamic methods. Static pruning approaches He et al. (2018); Li et al. (2017); Han et al. (2015); Castells et al. (2024); Ganjdanesh et al. (2024a) prune a pretrained model and use the pruned

model for all inputs. Conversely, dynamic pruning methods Elkerdawy et al. (2022); Kumar et al. (2024); Lin et al. (2020); Tang et al. (2021) employ a separate sub-network of the model for each input. Although static and dynamic pruning methods have been successful for image classification, they are not suitable to be directly applied to T2I models. Static pruning methods neglect prompt complexity and employ the same model for all prompts while dynamic pruning ideas cannot utilize batch-parallelism in GPUs. We refer to recent surveys White et al. (2023); He & Xiao (2023); Cheng et al. (2023) for a comprehensive review of pruning and NAS methods.

Our method distinguishes itself from existing approaches by introducing a prompt-based pruning technique for T2I models. This work marks the first instance where computational resources are allocated to prompts based on their individual complexities while ensuring optimal utilization and batch-parallelizability.

## C  OVERVIEW OF DIFFUSION MODELS

Given a random variable $\mathbf{x_0} \sim \mathcal{P}$, the goal of diffusion models Sohl-Dickstein et al. (2015); Ho et al. (2020) is to model the underlying distribution $\mathcal{P}$ using a training set $\mathcal{D} = \{x_0\}$ of samples. To do so, first, diffusion models define a forward process parameterized by $t$ in which they gradually perturb each sample $x_0$ with Gaussian noise with the variance schedule of $\beta_t$:

$$q(x_t|x_{t-1}) = \mathcal{N}(x_t; \sqrt{1-\beta_t}x_{t-1}, \beta_t I) \tag{16}$$

where $t \in [1, T]$. Thus, $q(x_t|x_0)$ has a Gaussian form:

$$q(x_t|x_0) = \mathcal{N}(x_t; \sqrt{\bar{\alpha}_t}x_0, (1-\bar{\alpha}_t)I) \tag{17}$$

where $\alpha_t = 1 - \beta_t$ and $\bar{\alpha}_t = \prod_{i=1}^{t} \alpha_i$. The noise schedule $\beta_t$ is usually selected Ho et al. (2020) such that $q(x_T) \to \mathcal{N}(0, I)$. Assuming $\beta_t$ is small, diffusion models approximate the denoising distribution $q(x_{t-1}|x_t)$ by a parameterized Gaussian distribution $p_\theta(x_{t-1}|x_t) = \mathcal{N}(x_{t-1}; \frac{1}{\sqrt{\alpha_t}}(x_t - \frac{\beta_t}{\sqrt{1-\bar{\alpha}_t}}\epsilon_\theta(x_t, t)), \sigma_t^2 I)$, and $\sigma_t^2$ is often set to $\beta_t$. Diffusion models implement $\epsilon_\theta(.)$ with a neural network called the denoising model and train it with the variational evidence lower bound (ELBO) objective Ho et al. (2020):

$$\mathcal{L}_{\text{DDPM}}(\theta) = \mathbb{E}_{\substack{t\sim[1,T] \\ \epsilon\sim\mathcal{N}(0,I) \\ x_t\sim q(x_t|x_0)}} ||\epsilon(x_t, t; \theta) - \epsilon||^2 \tag{18}$$

Similarly, T2I diffusion models train a denoising model using pairs of image and text prompts $(x_0, p) \sim \mathcal{P}$ to model the distribution $\mathcal{P}(\mathbf{x_0}|\mathbf{p})$ of images given an input text prompt:

$$\mathcal{L}_{\text{DDPM}}(\theta) = \mathbb{E}_{\substack{(x_0,p)\sim\mathcal{P} \\ t\sim[1,T] \\ \epsilon\sim\mathcal{N}(0,I) \\ x_t\sim q(x_t|x_0)}} ||\epsilon(x_t, p, t; \theta) - \epsilon||^2 \tag{19}$$

T2I Diffusion models generate a new sample by sampling an initial noise from $x_T \sim p(x_T) = \mathcal{N}(0, I)$ and iteratively denoising it using the denoising model by sampling from $p_\theta(x_{t-1}|x_t, p)$. Thus, the sampling process requires $T$ sequential forward calculation of the denoising model, making it a slow and costly process.

## D  MORE DETAILS OF APTP

In this section we provide more details of the architecture predictor, the prompt router module, and our pruning method.

## D.1 ARCHITECTURE PREDICTOR

$f_{\text{AP}}(\cdot)$ in Eq. 4 is the architecture predictor. The architecture predictor changes the dimensionality of $z$ to the dimensionality of $e$ and codes $a$, which is the number of prunable width and depth units in the T2I model. We implement the $f_{\text{AP}}$ function with a single feed-forward layer. Our prompt encoder, Sentence Transformer Reimers & Gurevych (2019), has an output dimension of **768**. The total number of prunable units in SD 2.1 is **1620**, so the architecture predictor has an input dimension of **768** and an output dimension of **1620** in all of our experiments.

## D.2 ROUTER MODULE DEFINITION

During the pruning process, we use the assignment matrix $Q^*$ calculated by optimal transport (Eq. 7) to route the architecture embeddings $e$ to the architecture codes $a$. Thus, during the pruning process, the definition of function $f_{\text{R}}$ is as follows:

$$I = \texttt{argmax}(BQ^*, \texttt{dim} = 0) \tag{20}$$

$$f_{\text{R}}(e,\ \mathcal{A},\ Q^*) = A[:, I] \tag{21}$$

where $e$ represents the architecture embeddings in a pruning batch, $\mathcal{A}$ is the set of architecture codes, $Q^*$ is the calculated optimal assignment matrix, and $A$ is the matrix of architecture codes.

At test time, the router function routes an input architecture embedding $e$ to the most similar trained architecture code:

$$f_{\text{R}}(e, \mathcal{A}) = \texttt{argmax}_{a \in \mathcal{A}} \frac{e^T a}{||e|| ||a||} \tag{22}$$

## D.3 PRUNING DEPTH

We apply the vectors $\mathbf{u}^{(i)} = [\mathbf{u}_j^{(i)}]_{j=1}^{M}$ (Eq. 9) to prune the model's depth with $M$ being the total number of layers we prune. As the U-Net Ronneberger et al. (2015) architecture of T2I models has skip connections, we prune the depth layers in the encoder and decoder sides separately.

**Pruning Depth in Encoder.** We prune the $j$-$th$ depth layer $f_j$ in the encoder by applying the vector $u_j^{(i)}$ with the following formulation:

$$\widehat{\mathcal{F}}_j = \mathbf{u}_j^{(i)} f_j(\mathcal{F}_{j-1}) + (1 - \mathbf{u}_j^{(i)})\mathcal{F}_{j-1} \tag{23}$$

where $\mathcal{F}_{j-1}$ is the feature maps of the previous layer. When $u_j^{(i)}$ is close to one/zero, the $j$-th layer will be kept/pruned.

**Pruning Depth in Decoder.** In the decoder, the input of each layer is a concatenation of feature maps of the previous layer and feature maps of its corresponding encoder layer from the skip connection $\mathcal{F}_{j,skip}$. Thus, we prune the $j$-th depth layer $f_j$ in the decoder as:

$$\widehat{\mathcal{F}}_j = \mathbf{u}_j^{(i)} f_j(\mathcal{F}_{j-1} || \mathcal{F}_{j,skip}) + (1 - \mathbf{u}_j^{(i)})\mathcal{F}_{j-1} \tag{24}$$

## D.4 DISTILLATION OBJECTIVE

Assuming $B^{(i)}$ samples get routed to the architecture code $a^{(i)}$ in a training batch ($\sum_i B^{(i)} = B$), we train the prompt router and the architecture codes using the following objective:

$$\min_{\eta, \mathcal{A}} \mathcal{L} = [\frac{1}{N} \sum_{i=1}^{N} [\frac{1}{B^{(i)}} \sum_{j=1}^{B^{(i)}} [\mathcal{L}_{\text{DDPM}}(x_j^{(i)}, p_j^{(i)}; a^{(i)}) + \lambda_{\text{distill}} \mathcal{L}_{\text{distill}}(x_j^{(i)}, p_j^{(i)}; a^{(i)})]]]$$
$$+ \lambda_{\text{res}} \mathcal{R}(\widehat{T}(\mathcal{A}), T_d) + \lambda_{\text{cont}} \mathcal{L}_{\text{cont}}(\eta) \tag{25}$$

$\mathcal{L}_{\text{DDPM}}(x_j^{(i)}, p_j^{(i)}; a^{(i)})$ denotes the denoising objective for the sample $(x_j^{(i)}, p_j^{(i)})$ routed to the sub-network chosen by the architecture code $a^{(i)}$ (Eq. 19). $\mathcal{R}(\widehat{T}(\mathcal{A}), T_d)$ regularizes the weighted average of the MACs used by architecture codes ($\widehat{T}(\mathcal{A}) = \sum_i \frac{B^{(i)}}{B} \widehat{T}(a^{(i)})$) to be close to $T_d$. We define

$$\mathcal{R}(x, y) = \log(\max(x, y)/\min(x, y)) \tag{26}$$

as it can keep the resource usage close to the target value. $\mathcal{L}_{\text{cont}}$, given by Eq. 13, is the contrastive loss that guides the architecture predictor to map prompt representations to the regions of the space of the architecture embeddings such that their corresponding architecture vectors maintain the similarity between the prompts. Finally, $\mathcal{L}_{\text{distill}}$ is the distillation objective Hinton et al. (2015), regularizing the pruned model to have similar outputs to the original one. We do distillation at two levels, output level and block level. The output level distillation objective is:

$$\mathcal{L}_{\text{output-distill}}(x_j^{(i)}, p_j^{(i)}; a^{(i)}) = \mathbb{E}_{\substack{t \sim [1,T] \\ \epsilon \sim \mathcal{N}(0,I) \\ x_{j,t}^{(i)} \sim q(x_t | x_j^{(i)})}} [||\epsilon_{\text{Teacher}}(x_{j,t}^{(i)}, p_j^{(i)}, t; \theta) - \epsilon_{\text{Sub-Net}}(x_{j,t}^{(i)}, p_j^{(i)}, t; \theta, a^{(i)})||^2] \tag{27}$$

Here, $\epsilon_{\text{Teacher}}(x_{j,t}^{(i)}, p_j^{(i)}, t; \theta)$ denotes the original model's output and $\epsilon_{\text{Sub-Net}}(x_{j,t}^{(i)}, p_j^{(i)}, t; \theta, a^{(i)})$ denotes the output of the sub-network chosen by the architecture code $a^{(i)}$. The way we prune the U-Net preserves the output shape of each block. Doing so enables us to do distillation at block level as well and regularize the sub-network to match the output of the original model at each block. The block-level distillation objective is:

$$\mathcal{L}_{\text{block-distill}}(x_j^{(i)}, p_j^{(i)}; a^{(i)}) = \mathbb{E}_{\substack{t \sim [1,T] \\ \epsilon \sim \mathcal{N}(0,I) \\ x_{j,t}^{(i)} \sim q(x_t | x_j^{(i)})}} [\sum_b ||\epsilon_{\text{Teacher}}^b(x_{j,t}^{(i)}, p_j^{(i)}, t; \theta) - \epsilon_{\text{Sub-Net}}^b(x_{j,t}^{(i)}, p_j^{(i)}, t; \theta, a^{(i)})||^2] \tag{28}$$

Here, $\epsilon_{\text{Teacher}}^b$ and $\epsilon_{\text{Sub-Net}}^b$ denote the outputs of block $b$ of the original model and the chosen sub-network of it, respectively. The total distillation loss $\mathcal{L}_{\text{distill}}$ is simply the sum of the output-level loss and the block-level distillation loss:

$$\mathcal{L}_{\text{distill}} = \mathcal{L}_{\text{output-distill}} + \mathcal{L}_{\text{block-distill}} \tag{29}$$

### D.5 Fine-tuning

The finetuning loss is a weighted average of the original DDPM objective (Eq. 19) and the distillation loss term (Eq. 29):

$$\mathcal{L}_{\text{finetuning}} = \alpha_{\text{DDPM}}\mathcal{L}_{\text{DDPM}} + \alpha_{\text{distill}}\mathcal{L}_{\text{distill}} \tag{30}$$

## E Experiments

### E.1 Models and Datasets

We use two datasets as our *target* datasets: Conceptual Captions 3M (CC3M) Sharma et al. (2018) and MS-COCO Captions 2014 Lin et al. (2014) with approximately 2.5M and 400K image-caption pairs, respectively. We apply APTP to the Stable Diffusion 2.1 (SD 2.1) Rombach et al. (2022) model. On CC3M, we prune SD2.1 with two settings: Base (0.85 budget, 16 experts) and Small (0.66 budget, 8 experts). Similarly, for COCO, we have two settings: Base (0.78 budget, 8 experts) and Small (0.64 budget, 8 experts). We use a pruned SD 2.1 using weight norm pruning Li et al. (2017) as a baseline for our main experiments.

### E.2 Experimental Settings

We train at a fixed resolution of $256 \times 256$ across all settings. During pruning, we first train the architecture predictor for 500 iterations as a warm-up phase. During this warm-up phase, we directly use its predicted architectures for pruning. Then, we start architecture codes and train the architecture predictor jointly with the codes for an additional 2500 iterations. We use the AdamW Loshchilov & Hutter (2019) optimizer and a constant learning rate of 0.0002 for both modules, with a 100-iteration linear warm-up. The effective pruning batch size is 1024, achieved by training on 16 NVIDIA A100 GPUs with a local batch size of 64. The temperature of the Gumbel-Sigmoid reparametrization (Eq. 9) is set to $\gamma = 0.4$. We set the regularization strength of the optimal transport objective (Eq. 5) to $\epsilon = 0.05$. We use 3 iterations of the Sinkhorn-Knopp algorithm Cuturi (2013) to solve the optimal transport problem Caron et al. (2020). We set the contrastive loss temperature $\tau$ to 0.03. The total pruning loss is the weighted average of DDPM loss, distillation loss, resource loss, and contrastive loss (see Eq. 15) with weights $\lambda_{\text{distill}} = 0.2$, $\lambda_{\text{res}} = 2.0$, and $\lambda_{\text{cont}} = 100.0$. After the pruning phase, we fine-tune the experts with the prompts assigned to them for 30,000 iterations using the AdamW optimizer, a fixed learning rate of 0.00001, and a batch size of 128. Upon experiments, we observed that higher weights of the DDPM loss result in unstable fine-tuning and slow convergence. As a result, we set the DDPM loss weight in the fine-tuning loss (Eq. 30) $\alpha_{\text{DDPM}}$ to 0.0001. We set $\alpha_{\text{distill}} = 1.0$. For sample generation, we use the classifier-free guidance Ho & Salimans (2022) technique with the scale of 7.5 and 25 steps of the PNDM sampler Liu et al. (2022).

### E.3 Evaluation

For quantitative evaluation of models pruned on CC3M, we use its validation dataset of approximately 14k samples. For COCO, we sample 30k captions of unique images from its 2014 validation dataset. We report Fréchet inception distance (FID) Heusel et al. (2017), CLIP score Hessel et al. (2021), and Maximum Mean Discrepancy with CLIP Embeddings (CMMD) Jayasumana et al. (2023) for APTP, the baselines, and SD 2.1 itself.

### E.4 Results

#### E.4.1 Training Loss

Introduced in section 3.3.1, the resource loss regularizes the weighted average of the MACs used by architecture codes ($\widehat{T}(\mathcal{A}) = \sum_i \frac{B^{(i)}}{B} [\widehat{T}(a^{(i)})]$) to be close to $T_d$. We define resource loss as:

$$\mathcal{R}(x, y) = \log(\max(x, y)/\min(x, y)) \tag{31}$$

Fig. 6a illustrates the resource loss when applying APTP-Base to prune Stable Diffusion 2.1 using the COCO dataset as the target. This is shown under two conditions: with and without optimal transport following the initial warm-up phase (refer to E for details). APTP effectively regularizes the model so average MACs of the architecture codes is very close to the target budget. Fig. 6b presents the contrastive loss (Eq. 13) under the same conditions. APTP maps the prompt embeddings to regions of architecture embeddings such that their corresponding architectures maintain the similarity between prompts.

#### E.4.2 Visualization of the Impact of Optimal Transport

Fig. 7 displays the assignment of prompts to experts with and without optimal transport. By adding optimal transport, the assignment becomes more diverse, ensuring all experts get enough samples. This results in a significant improvement of performance metrics (See Table 3). Fig. 8 shows the distribution of the number of training CC3M samples mapped to each expert of APTP-Base.

#### E.4.3 Analysis of Prompt Router on COCO

Table 4 demonstrates the most frequent words in the prompts assigned to each expert of APTP-Base pruned on COCO, revealing distinct topics and effective specialization. For example, Expert 1 specializes in indoor scences, Expert 5 in wildlife, and Expert 6 in urban scences. Expert 8 which has the highest resource utilization, focuses on images of human beings and hands, an observation

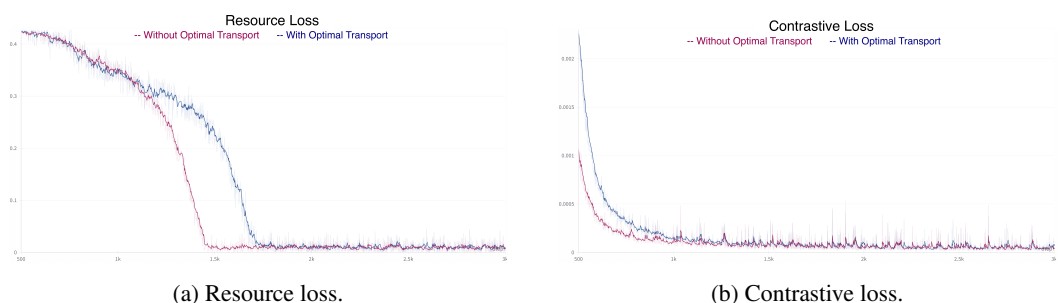

(a) Resource loss.                                    (b) Contrastive loss.

Figure 6: Resource and Contrastive loss observed when applying APTP-Base with a MAC budget of 0.77 to prune Stable Diffusion 2.1 using the COCO dataset. The comparison is made between two settings: with and without optimal transport. APTP both adheres to the target MAC budget and finds architecture vectors that maintain the similarity between the prompts.

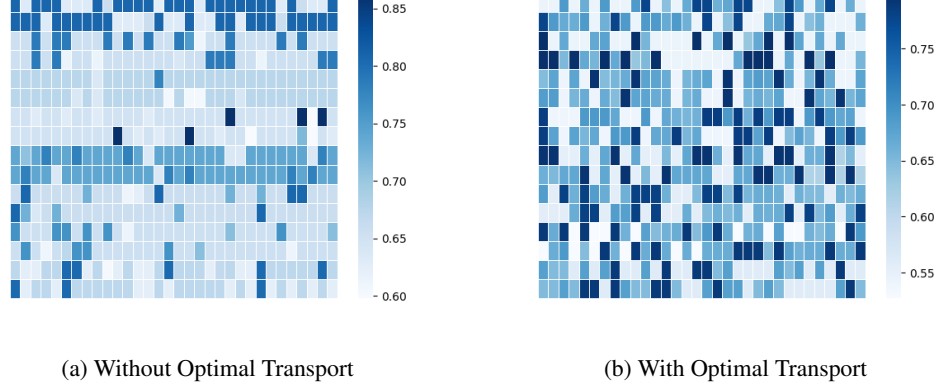

(a) Without Optimal Transport                    (b) With Optimal Transport

Figure 7: Comparison of sample assignments in a batch to experts with and without optimal transport. The incorporation of optimal transport results in a more diverse assignment pattern. In the figure, each square represents a prompt within the batch, and the color signifies the budget level of the expert assigned to the prompt. Higher-resource experts are indicated by darker blue.

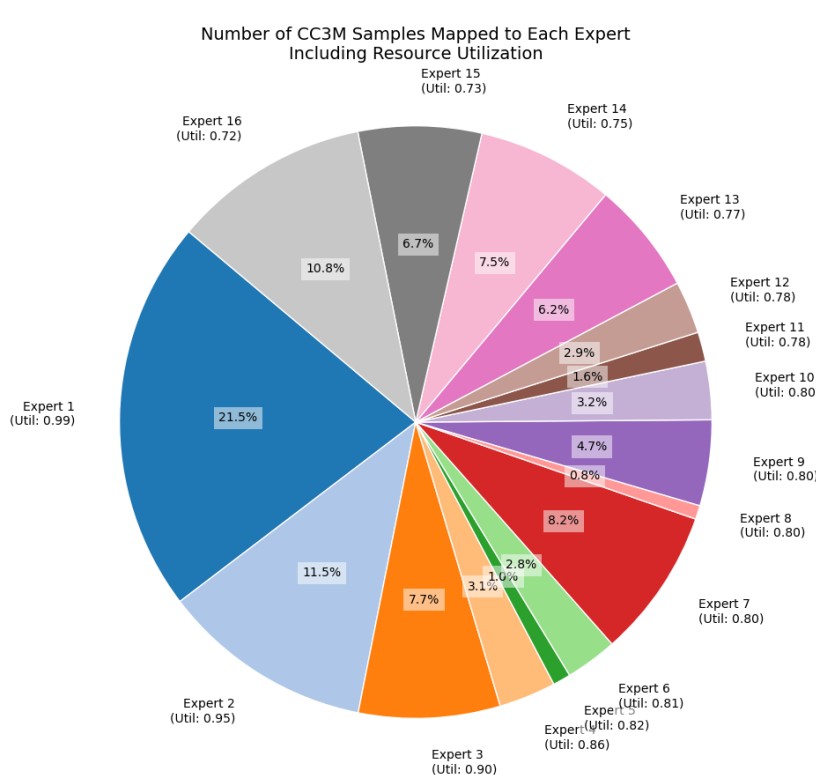

Figure 8: Distribution of CC3M Samples Mapped to Each Expert of APTP-Base, Including Resource Utilization Ratios

consistent with our prompt analysis on CC3M (Table 2). Hands are another category that have been found to be challenging for SD 2.1 Gandikota et al. (2023).

Table 4: The most frequent words in prompts assigned to each expert of APTP-Base pruned on COCO. The resource utilization of each expert is indicated in parentheses.

| Expert 1 (0.65, Indoor Scenes and Dining) | Expert 2 (0.77, Food and Small Groups) |
|---|---|
| table - plate - kitchen - sitting | food - pizza - sandwich |
| **Expert 3 (0.78, People and Objects)** | **Expert 4 (0.79, Sports and Activities)** |
| skateboard - surfboard - laptop - tie - phone | tennis - baseball - racquet - skateboard - skis |
| **Expert 5 (0.79, Wildlife and Nature)** | **Expert 6 (0.80, Urban Scenes and Transportation)** |
| giraffe - herd - sheep - zebra - elephants | street - train - bus - park - building |
| **Expert 7 (0.81, Outdoor Activities and Nature)** | **Expert 8 (0.83, Domestic Life and Pets)** |
| beach - ocean - surfboard - kite - wave | man - woman - girl - hand - bed - cat |

### E.4.4 QUANTITATIVE RESULTS

The quantitative results on the CC3M and MS-COCO datasets (Table 5) indicate that APTP significantly outperforms the baseline Norm method in terms of FID and CLIP scores while also reducing complexity. For both datasets, APTP at 30k and 50k fine-tuning iterations shows lower FID and CMMD values and higher CLIP scores compared to the baseline, demonstrating the advantage of prompt based pruning compared to static pruning ideas for T2I models.

We also benchmark our method and the baselins on the PartiPrompt Yu et al. (2022) dataset using PickScore Kirstain et al. (2023). The prompts in this benchmark can be considered out-of-distribution for our model as many of them are significantly longer and semantically different from the ones in MSCOCO. We report the PickScore Kirstain et al. (2023) as a proxy for human preference and present the results in Table 2 below. We can observe that the Pickscore of the pruned model is only 1% below the original Stable Diffusion 2.1, indicating that the pruned model can preserve the general knowledge and generation capabilities of the Stable Diffusion model. Table 6 demonstrates the results.

Conducting systematic, controlled evaluations of large-scale text-to-image models remains challenging, as most existing models, datasets, or implementations are not publicly available. Training a new model from scratch is prohibitively expensive, even when training code is accessible. Extending the design decision of architecture design methods to other compute budgets or settings is highly non-trivial. In contrast, our approach can be applied in a plug-and-play manner to any target model capacity, delivering competitive performance with drastically reduced hardware compute time.

In Table 7, we compare our model to recent state-of-the-art text-to-image models, based on available information and their reported values, acknowledging significant differences in training datasets, iteration counts, batch sizes, and model sizes. We also include a column detailing the compute used to train these models, with estimates derived from the papers and public sources. Our findings demonstrate that APTP achieves competitive results post-training while requiring several orders of magnitude less compute time.

### E.4.5 EXPERT ARCHITECTURES

Figures 9 and 10 display the block-level U-Net architecture of all experts of APTP-Base, with CC3M and COCO as the target datasets, respectively. The retained MAC patterns are markedly different, further corroborating that different datasets require distinct architectures and that a 'one-size-fits-all' approach is not well-suited for T2I models. For CC3M, down-sampling blocks generally are retained with higher ratios. Overall, APTP tends to prune Resnet Blocks more frequently and aggressively compared to Attention Blocks.

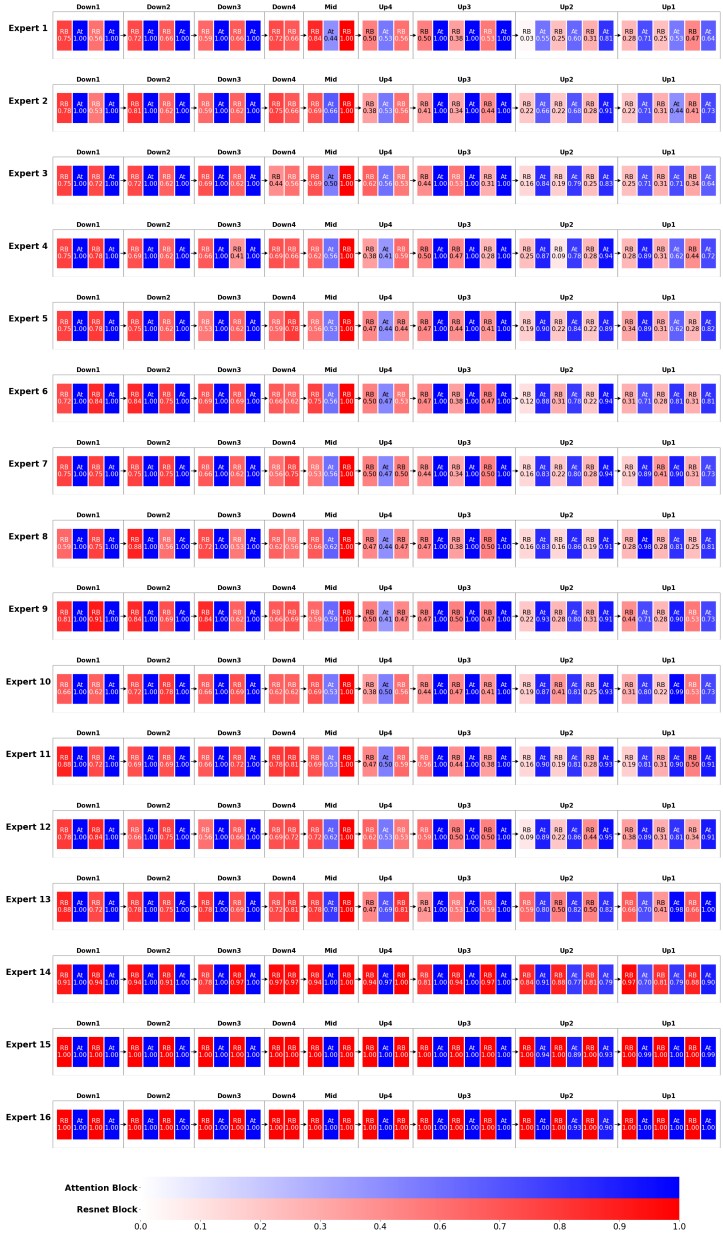

Figure 9: The block-level retained MAC ratio of the UNet architecture of all experts of APTP-Base applied to Stable Diffusion 2.1 with CC3M as the target dataset. The groups of ResBlocks and the heads of Attention Blocks are pruned based on the outputs of the architecture predictor. The intensity of the color of each block represents the resource utilization of it. The number in each block indicates the precise ratio of retained MACs of the block. Conv_in, Conv_out, and skip connections between corresponding down and up blocks are omitted for brevity.

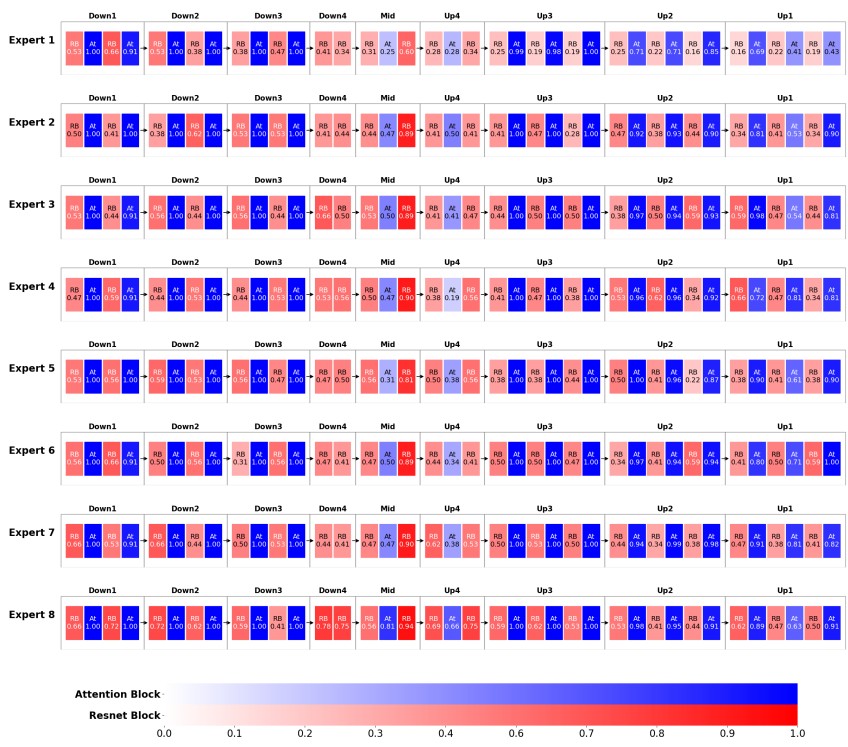

Figure 10: The block-level retained MAC ratio of the UNet architecture of all experts of APTP-Base applied to Stable Diffusion 2.1 with COCO as the target dataset. The groups of ResBlocks and the heads of Attention Blocks are pruned based on the outputs of the architecture predictor. The intensity of the color of each block represents the resource utilization of it. The number in each block indicates the precise ratio of retained MACs of the block. Conv_in, Conv_out, and skip connections between corresponding down and up blocks are omitted for brevity.

Table 5: Quantitative results on CC3M and MS-COCO. We report the performance metrics using samples generated at the resolution of 768 and downsampled to 256 Kim et al. (2023). We measure models' MACs and Latency with the input resolution of 768 on an A100 GPU. @30/50k shows the model's fine-tuning iterations after pruning.

CC3M

| Method | Complexity | | Performance | | |
| | MACs (@768) | Latency (↓) (Sec/Sample) (@768) | FID (↓) | CLIP (↑) | CMMD (↓) |
|---|---|---|---|---|---|
| Norm Li et al. (2017) @30k | 1185.3G | 3.4 | 157.51 | 26.23 | 1.778 |
| Norm Li et al. (2017) @50k | 1185.3G | 3.4 | 141.04 | 26.51 | 1.646 |
| APTP(0.66) @30k | 916.3G | 2.6 | 60.04 | 28.64 | 1.094 |
| APTP(0.66) @50k | 916.3G | 2.6 | 54.95 | 29.08 | 1.017 |
| APTP(0.85) @30k | 1182.8G | 3.4 | 36.77 | 30.84 | 0.675 |
| APTP(0.85) @50k | 1182.8G | 3.4 | 36.09 | 30.90 | 0.669 |
| SD 2.1 | 1384.2G | 4.0 | 32.08 | 31.12 | 0.567 |

(a)

MS-COCO

| Method | Complexity | | Performance | | |
| | MACs (@768) | Latency (↓) (Sec/Sample) (@768) | FID (↓) | CLIP (↑) | CMMD (↓) |
|---|---|---|---|---|---|
| Norm Li et al. (2017) @30k | 1077.4G | 3.1 | 60.42 | 27.06 | 1.524 |
| Norm Li et al. (2017) @50k | 1077.4G | 3.1 | 47.35 | 28.51 | 1.136 |
| APTP(0.64) @30k | 890.0G | 2.5 | 39.12 | 29.98 | 0.867 |
| APTP(0.64) @50k | 890.0G | 2.5 | 36.17 | 30.21 | 0.739 |
| APTP(0.78) @30k | 1076.6G | 3.1 | 22.60 | 31.32 | 0.569 |
| APTP(0.78) @50k | 1076.6G | 3.1 | 22.26 | 31.38 | 0.561 |
| SD 2.1 | 1384.2G | 4.0 | 15.47 | 31.33 | 0.500 |

(b)

Table 6: Results on PartiPrompts. We report performance metrics using samples generated at the resolution of 768. We measure models' MACs/Latency with the input resolution of 768 on an A100 GPU. @30/50k shows fine-tuning iterations after pruning.

Train on CC3M

| Method | Complexity | | PartiPrompts |
| | MACs (@768) | Latency (↓) (Sec/Sample) (@768) | PickScore (↑) |
|---|---|---|---|
| Norm (Li et al., 2017) @50k | 1185.3G | 3.4 | 18.114 |
| SP (Fang et al., 2023) @30k | 1192.1G | 3.5 | 18.727 |
| BKSDM (Kim et al., 2023) @30k | 1180.0G | 3.3 | 19.491 |
| APTP(0.66) @30k | 916.3G | 2.6 | 19.597 |
| APTP(0.85) @30k | 1182.8G | 3.4 | 21.049 |
| SD 2.1 | 1384.2G | 4.0 | 21.316 |

(a)

Train on MS-COCO

| Method | Complexity | | PartiPrompts |
| | MACs (@768) | Latency (↓) (Sec/Sample) (@768) | PickScore (↑) |
|---|---|---|---|
| Norm (Li et al., 2017) @50k | 1077.4G | 3.1 | 18.563 |
| SP (Fang et al., 2023) @30k | 1071.4G | 3.3 | 19.317 |
| BKSDM (Kim et al., 2023) @30k | 1085.4G | 3.1 | 19.941 |
| APTP(0.64) @30k | 890.0G | 2.5 | 20.626 |
| APTP(0.78) @30k | 1076.6G | 3.1 | 21.150 |
| SD 2.1 | 1384.2G | 4.0 | 21.316 |

(b)

Table 7: Comparison of APTP and SOTA Text-to-Image architecture design methods.

| Models | Type | Sampling | Steps | FID-30K(↓) | CLIP(↑) | Params (B) | Images (B) | Compute (GPU/TPU days) |
|---|---|---|---|---|---|---|---|---|
| GigaGAN (Kang et al., 2023) | GAN | 1-step | 1 | 9.09 | - | 0.9 | 0.98 | 4783 A100 |
| Cogview-2 (Ding et al., 2022) | AR | 1-step | 1 | 24.0 | - | 6.0 | 0.03 | - |
| DALLE-2 (Ramesh et al., 2022) | Diffusion | DDPM | 292 | 10.39 | - | 5.20 | 0.25 | 41667 A100 |
| Imagen (Saharia et al., 2022) | Diffusion | DDPM | 256 | 7.27 | - | 3.60 | 0.45 | 7132 TPU |
| SD2.1 (Rombach et al., 2022) | Diffusion | DDIM | 50 | 9.62 | 0.304 | 0.86 | > 2 | 8334 A100 |
| Pixart-$\alpha$ Chen et al. (2023) | Diffusion | DDPM | 20 | 10.65 | - | 0.6 | 0.025 | 753 A100 |
| SnapFusion (Li et al., 2024) | Diffusion | Distilled | 8 | 13.5 | 0.308 | 0.85 | - | > 128 A100 |
| MobileDiffusion (Zhao et al., 2023) | Diffusion | DDIM | 50 | 8.65 | 0.325 | 0.39 | 0.15 | 7680 TPU |
| RAPHAEL (Xue et al., 2023) | Diffusion | DDPM | - | 6.61 | - | 3.0 | > 5 | 60000 A100 |
| APTP-Base (@30k) | Diffusion | DDIM | 50 | 19.14 | 0.318 | 0.65 | 0.0001 | 6.5 A100 |

### E.4.6 MORE VISUAL RESULTS

Table 8 displays the prompts for the images in Fig. 3 from CC3M and COCO validation sets.

| **CC3M Prompts** |
| --- |
| Saw this beautiful sky on my way home. |
| A silhouetted palm tree with boat tied to it rests on a beach that a man walks across during golden hour. |
| Sketch of a retro photo camera drawn by hand on a white background. |
| From left: person person, the dress on display. |
| Never saw a doll like this before but she sure is sweet looking. |
| The team on the summit. |
| A water drop falls towards a splash already made by another water drop. |
| Husky dog in a new year's interior. |
| Old paper with a picture of flowers, ranked in a moist environment. |
| People on new year's eve! |
| I drive over stuff - a pretty cool jeep, 4x4, or truck t-shirt. |
| The crowds arrive for day of festival. |
| A scary abandoned house under a starry sky. |
| Freehand fashion illustration with a lady with decorative hair. |
| Introduce some new flavors to your favorite finger food with these inspired chicken wings. |
| Gloomy face of a sad woman looking down, zoom in, gray background. |

| **COCO Prompts** |
| --- |
| A white plate topped with a piece of chocolate covered cake. |
| Decorated coffee cup and knife sitting on a patterned surface. |
| A desk topped with a laptop computer and speakers. |
| A man sitting on the beach behind his surfboard. |
| A pizza type dish with vegetables and a lemon wedge. |
| The browned cracked crust of a baked berry pie. |
| A tennis player in an orange skirt walks off the court. |
| The skier in the helmet moves through thick snow. |
| A giraffe walks leisurely through the tall grass. |
| Several brown horses are standing in a field. |
| A red fire hydrant on a concrete block. |
| A bus driving in a city area with traffic signs. |
| There is a man on a surf board in the ocean. |
| A boat parked on top of a beach in crystal blue water. |
| A small kitchen with low a ceiling. |
| A calico cat curls up inside a bowl to sleep. |

Table 8: Prompts for Fig. 3

We also provide more samples from APTP-Base pruned on CC3M and COCO. Fig. 11 presents samples from the validation set of CC3M generated by each 16 experts of APTP-Base at $0.85$ MACs budget. Fig. 12 shows samples from the validation set of COCO from each of the 8 experts of APTP-Base pruned to $0.78$ MACs budget.

We provide samples from APTP pruned models on various styles in Fig. 13. APTP preserves the original model's versatility and generalizes well to out-of-distribution prompts.

We provide samples comparing APTP to the best baseline, namely BKSDM Kim et al. (2023)), on different prompts from the PartiPrompts Yu et al. (2022) datasets in Fig. 14. APTP produces better images compared to the baseline.

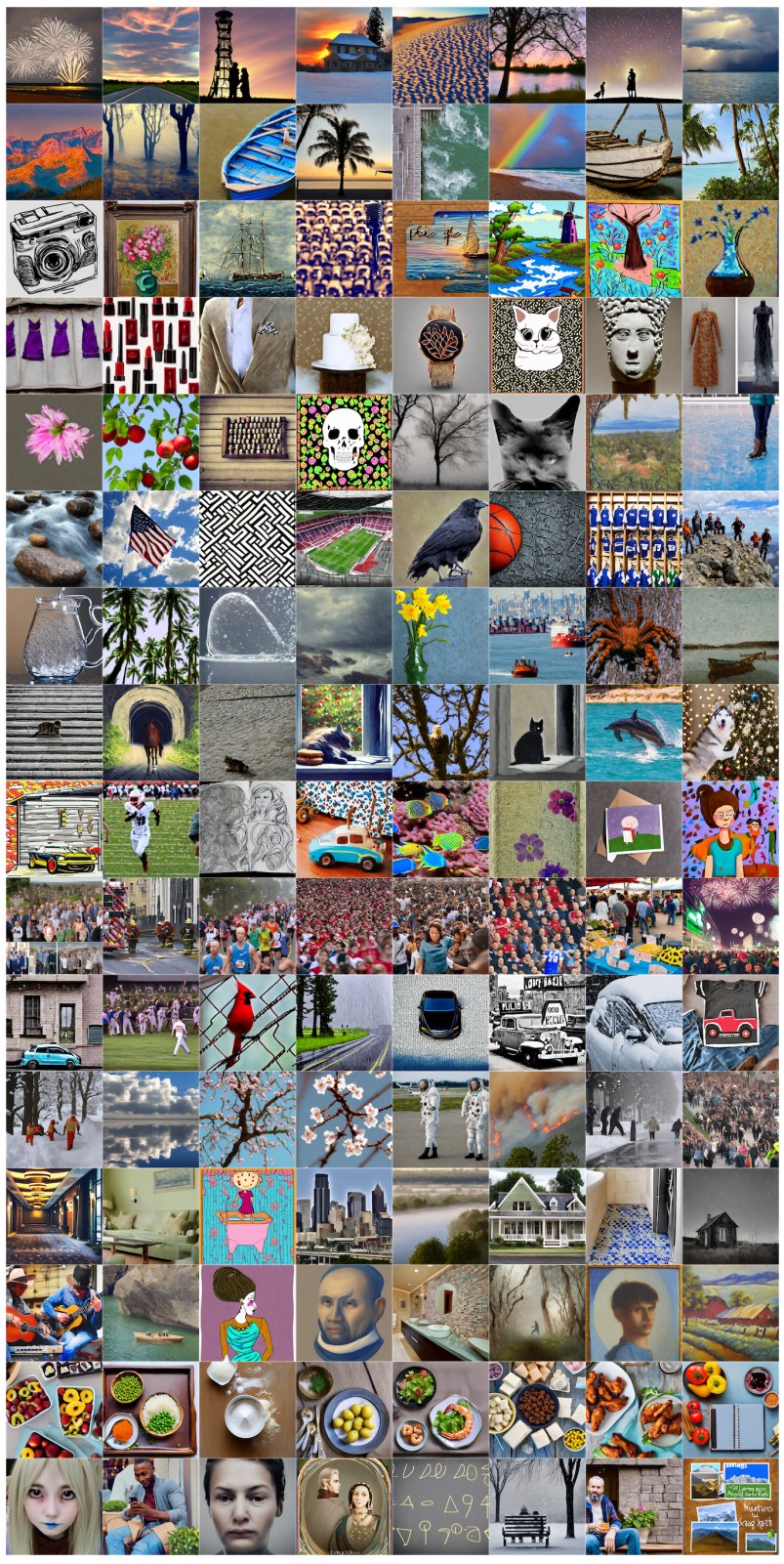

Figure 11: Samples of the APTP-Base experts after pruning the Stable Diffusion V2.1 using CC3M Sharma et al. (2018) as the *target* dataset. Each row corresponds to a unique expert. Please refer to Table 2 for the groups of prompts assigned to each expert.

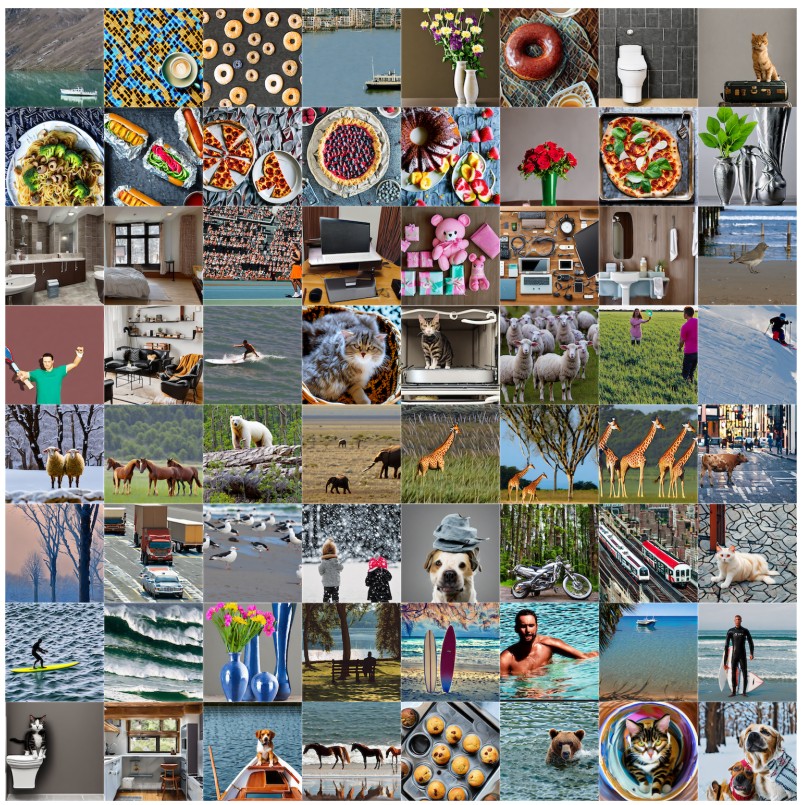

Figure 12: Samples of the APTP-Base experts after pruning the Stable Diffusion V2.1 using MS-COCO Lin et al. (2014) as the *target* dataset. Each row corresponds to a unique expert. Please refer to Table 4 for the groups of prompts assigned to each expert.

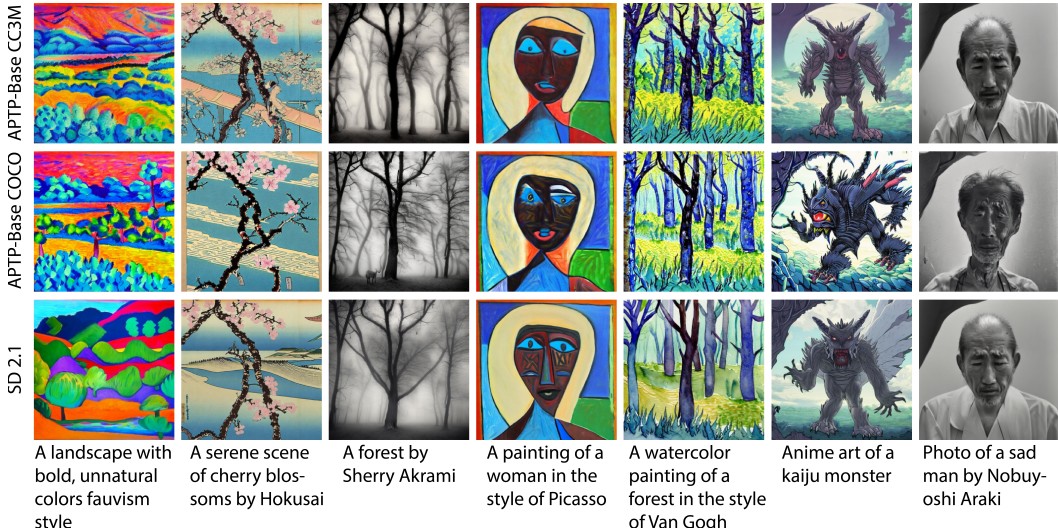

Figure 13: APTP generalizes to various styles, even if they are not present in the target dataset.

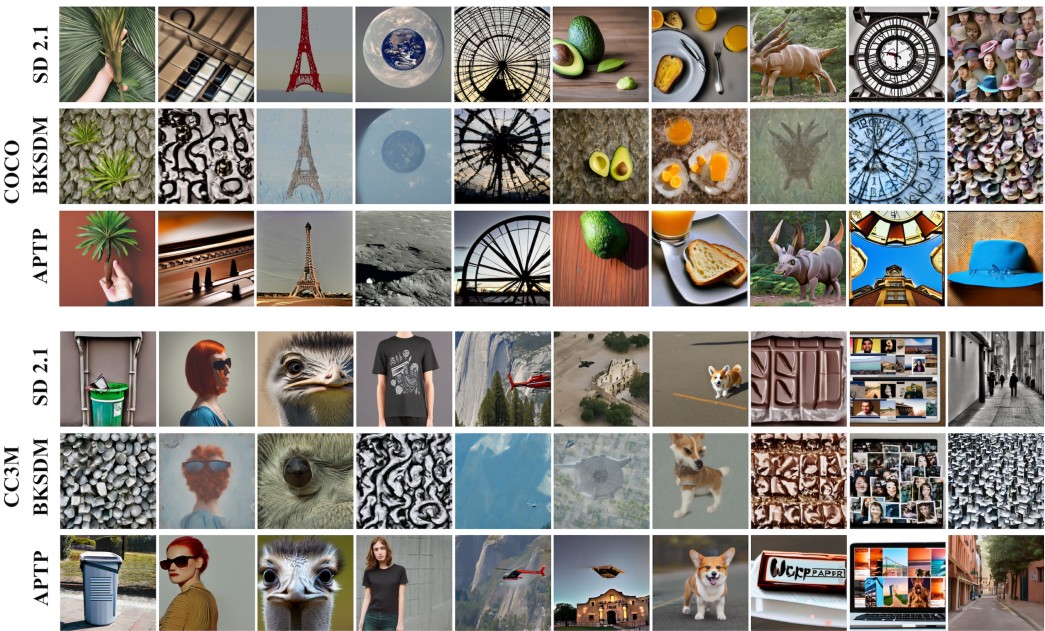

Figure 14: A visual comparison of randomly sampled generated images from Stable Diffusion 2.1, BKSDM (the best baseline), and APTP. The top section displays models trained on the COCO dataset, while the bottom section showcases samples from models trained on CC3M. Overall, APTP produces higher-quality images compared to the baseline. For prompts, see Table 9.

| Prompts |
| --- |
| a handpalm with leaves growing from it |
| a close-up of the keys of a piano |
| the Eiffel Tower in a desert |
| a view of the Earth from the moon |
| the silhouette of the Milllenium Wheel at dusk |
| an avocado on a table |
| a glass of orange juice to the right of a plate with buttered toast on it |
| a Styracosaurus |
| view of a clock tower from below |
| a hat |
| a trash bin |
| a woman with sunglasses and red hair |
| a close-up of an ostrich's face |
| a black t-shirt |
| A helicopter flies over Yosemite. |
| a spaceship hovering over The Alamo |
| a corgi |
| A bar of chocolate without a wrapper that has the word "WRAPPER" printed on it. |
| a laptop screen showing a bunch of photographs |
| a street |

Table 9: Prompts for Fig. 14

