# OpenReview forum: "Not All Prompts Are Made Equal: Prompt-based Pruning of Text-to-Image Diffusion Models"
_ICLR.cc/2025/Conference — ICLR 2025 Poster_

### Official Review · Reviewer_xWxf · 2024-11-01

**Soundness:** 3
**Presentation:** 3
**Contribution:** 3
**Rating:** 8
**Confidence:** 4

**Summary:**

The authors propose a mixture-of-expert-esque strategy for efficiently, which they coin Adaptive Prompt Tailored Pruning (APTP). The methods combine the benefits of dynamic and static pruning methods and archives good generative metrics while decreasing the computational cost.

**Strengths:**

* Good Illustrations and concise explanation
* The core idea is an interesting combination of known concepts in a new, but highly relevant setup.
* A good amount of comparison to other methods using multiple datasets with different image-types (COCO being primarily real-world images and CC3M being more diverse)

**Weaknesses:**

* While FID, CLIP-Score and CMMD alongside the visual examples provide a good overview, I, personally, would have preferred some human-user study (which I see is a lot of effort and for this reason, would not ask from the authors). As an alternative substitute, I propose compute the pick-scores [Kirstain et al. 2023] against the pruning metrics similar to [Pernias et al 2024] on a set of diverse prompts like Partiprompts could provide additional, easily interpretable evidence of this methods' superiority in terms of generation quality.

Kirstain et al. 2023 https://proceedings.neurips.cc/paper_files/paper/2023/hash/73aacd8b3b05b4b503d58310b523553c-Abstract-Conference.html
[Pernias et al. 2024] https://proceedings.neurips.cc/paper_files/paper/2023/hash/73aacd8b3b05b4b503d58310b523553c-Abstract-Conference.html

**Questions:**

* I would be curious to know how big the router-model itself is (in terms of parameters and memory footprint) and, by extension, how does the size of the router model affect the maximum batch size on an A100 GPU?
* Did you do any additional experiments concerning varying resolutions and aspect ratios and their impact on the pruned-image quality?
* Did you try applying this technique to transformer-based image-generation models like Pix-Art-Alpha / Sigma, do you see any major hurdles regarding the switch from ConvNets to Transformers?
* How specific is the APTP-model to its training data? How do out-of-distribution prompts impact the quality of the generations?

---

> ### Author Response · Authors · 2024-11-21
> **Response to Reviewer xWxf**
>
> We sincerely thank Reviewer xWxf for their thoughtful and constructive feedback, as well as for recognizing the strengths of our work, including its clear presentation, innovative combination of concepts, and extensive comparisons across datasets. Below, we address the specific concerns, questions, and suggestions raised.
>
> ## 1. PickScores and Evaluation on PartiPrompts
> Please see **section 2 of the global comment**.
> We appreciate the suggestion to use PickScores for evaluation. As noted in the global response, we have computed PickScores following Kirstain et al. (2023) on the PartiPrompts benchmark. Our results indicate that the pruned model achieves PickScores close to the original model, demonstrating minimal perceptual quality degradation and strong generalization capabilities. These results support the effectiveness of APTP in maintaining generation quality even for prompts beyond its training distribution.
>
> ## 2. Router Model Size and Impact
>
> The router model in our framework is based on a sentence transformer, with approximately 105M parameters. To make it more suitable for prompt routing, we added a lightweight linear layer on top of it.
> In terms of memory and computational efficiency:
>   1. The sentence transformer processes $\sim$2,800 prompts per second on a single GPU [1], contributing a negligible latency of $\sim$0.0003 seconds per prompt.
>
>   2. Its memory footprint is small compared to the main generative model, ensuring that it does not significantly impact the maximum batch size on an A100 GPU.
>
> While our experiments show that the sentence transformer is sufficient for prompt routing, the APTP framework is flexible and can integrate more advanced models like T5 or larger language models if needed. This flexibility allows APTP to adapt to more complex routing requirements in future use cases.
>
> ## 3. Higher Resolution and Aspect Ratios
>
> We did not evaluate any experiment at higher resolutions as we did not have the resources for doing all experiments in higher resolution to have a fair comparison. However, in response to the reviewer’s request, we conducted an additional experiment at 512×512 resolution on MSCOCO with only 5k iterations. The resulting FID score is 24.74, which demonstrates  that the effectiveness of APTP scales across resolutions.
>
> We leave a more comprehensive study of varying resolutions and aspect ratios for future work.
>
> ## 4. Application to Transformer-Based Architectures
> Please see **global comment section 3**.
>
> As the reviewer requested, we evaluated APTP on Stable Diffusion 3 (MM-DiT) model with 2 billion parameters, trained using a rectified flow objective. APTP achieved a latency reduction of $30$% per sampling step, while keeping the FID/CLIP scores on MSCOCO comparable to original model's performance. These results underscore the generality of our framework across architectures. Importantly, APTP preserves the Pickscore, a proxy for human preference, on Partiprompts, an out-of-distribution benchmark, within $98$% of the original model.
>
> ## 5- Out of Distribution Generalization
> We tested the APTP-pruned model as well as the baselines against out-of-distribution prompts from the PartiPrompts benchmark. Despite the distinct differences in prompt length, structure and semantics, our model achieved a PickScore within 99% of the original Stable Diffusion model.
>
> Moreover, qualitative visualizations on out-of-distribution prompts from various styles (Figure 6 at the beginning of the Appendix) show that APTP effectively preserves the diversity and fidelity of generated images. These results confirm the robustness of the pruned models to prompts beyond the training data distribution.

---

### Official Review · Reviewer_vGxh · 2024-11-02

**Soundness:** 3
**Presentation:** 2
**Contribution:** 1
**Rating:** 6
**Confidence:** 4

**Summary:**

This paper presents a new approach for accelerating the sampling from diffusion models using adaptive pruning denoted as APTP. The method is tailored for a specific scenario where the inference is on a specific known data distribution and the data for it is given (e.g. a company’s private dataset). Using this data, APTP trains a prompt router module that selects which parts of the architecture should be pruned for each given prompt. The selection is from a fixed set of reduced architecture states (denoted as architecture codes). Both the prompt router and arch. codes are trained end-to-end.

**Strengths:**

- The idea of pruning different parts of the network for each prompt is non-trivial and interesting.
- Visual results show APTP seems to use various pruning modes, and does not collapse to a single one.

**Weaknesses:**

**(1) Missing Baselines of Less Diffusion Steps:** My main concern is that the paper does not compare to approaches that perform less diffusion steps. More specifically the following should be considered as baselines:

- *Step Skipping Distillation of Diffusion Models [1,2,3]:* As mentioned in the paper, several methods tackled faster inference of diffusion models using this idea, mostly by knowledge distillation of the model to inference with fewer (between 1-4) steps. These methods should cut the latency by 8-25 times from the original model, while the proposed APTP was shown to cut latency by much less (not even 50% of the original model timing). These approaches also require training as APTP does, and their weights are readily available for many SoTA architectures, e.g. SDXL, SD 1.5, etc.

- *Caching Based Acceleration [4]:* These approaches cache features of the U-Net architecture and reuse them in later steps. Such approaches hold a distinct advantage of being training-free.

- *Less Denoising Steps at Inference Time:* A trivial training-free baseline is to simply sample images with less denoising steps. I wonder how that would compare to the proposed APTP. Can APTP further accelerate such a setting?

As step skipping approaches became more popular recently, I believe including some of these as baselines is a must for such an approach.

**(2) Quality of Writing (Sec. 3):** Sec. 3 is complicated and difficult to understand. Specifically, I think Sec. 3.2,3.3 are overloaded and could be shortened to a much clearer version. These subsections are the only place in the paper that describe the actual approach of APTP (and what does arch. codes mean), therefore are especially important for the proposed approach.

**(3) Visual Comparisons:** The paper offers a very limited selection of visual comparisons - having only 8 qualitative comparisons to the original baseline, and no such comparisons to previous approaches. Could the authors please supply a larger set of comparisons to the original model and baselines (including a step skipping distillation [1,2,3], caching [4] and less denoising steps).

**(4) Clustering of Pruning Modes:** While this qualitative analysis was promised in the abstract and introduction, it only exists in figures at the last 2 pages of the appendix without any textual description. Given it is mentioned in the abstract I think it should be included as a part of the main manuscript.

**(5) Limited Setting and Empirical Results:** Unlike other approaches, the proposed method is limited to a specific data distribution. Although the task is much more confined, the reduction in latency is not substantial: To keep performance comparable to the original model in terms of FID or CLIP score, APTP can only reduce 20% of the latency (Tab.1).

[1] Liu, Xingchao, et al. "Instaflow: One step is enough for high-quality diffusion-based text-to-image generation." The Twelfth International Conference on Learning Representations. 2023.

[2] Sauer, Axel, et al. "Adversarial diffusion distillation." European Conference on Computer Vision. Springer, Cham, 2025.

[3] Meng, Chenlin, et al. "On distillation of guided diffusion models." Proceedings of the IEEE/CVF Conference on Computer Vision and Pattern Recognition. 2023.

[4] Ma, Xinyin, Gongfan Fang, and Xinchao Wang. "Deepcache: Accelerating diffusion models for free." Proceedings of the IEEE/CVF Conference on Computer Vision and Pattern Recognition. 2024.

**Questions:**

- Did the authors try their approach on other architectures? even other backbones of Stable Diffusion?

---

> ### Author Response · Authors · 2024-11-21
> **Response to Reviewer vGxh**
>
> We thank the reviewer for their feedback and for finding our approach non-trivial and interesting. We address the concerns raised below and reiterate key points discussed in the general comment section.
>
> ## 1. Comparison with Few/One-Step Generation using step distillation and Caching Methods
> Please see  **global comment 1.2**.
>
> To clarify any confusion, the latency values in Table 1 reflect improvements for a single model evaluation, comparing the APTP-pruned model to the original Stable Diffusion model with the same number of sampling steps. Combining APTP with methods that reduce the number of evaluations (e.g., fewer sampling steps) would yield even greater efficiency gains. For example, step distillation methods could be applied during fine-tuning of pruned models, or step-caching methods could be used on the final APTP-pruned model. Currently, we use a combination of DDPM and distillation loss, but any diffusion training or distillation recipe could be used during fine-tuning.
>
> ## 2. Quality of writing
> We sincerely thank the reviewer for pointing this out. In response, we have added a concise and clear overview of the routing module to section 3.2. We believe this improves the clarity of the section and enhances understanding of the notations and equations. Please let us know if further clarification is needed, and we will be happy to revise it.
>
> ## 3. Visual Comparison
>  While APTP is not directly comparable to step-skipping methods (as discussed in the Related Work (L817-823 of the original and L925-930 in the revised pdf) and **section 1.2 of the global comment**), we acknowledge that including more visual comparisons with baselines strengthens our work, and we appreciate this suggestion. To address this, we have now included a new figure (Figure 7 currently at the beginning of the appendix for easy reference) showing outputs from the original Stable Diffusion model, the APTP-pruned model, and BKSDM as the best baseline method across various concepts. We have also added Figure 6 which compares the outputs of APTP-pruned model and SD2.1 on various styles. Figure 6 and 7 demonstrate that the APTP-pruned model generalizes well to concepts not present in the target dataset while outperforming baselines in image generation.
>
> ## 4. Clustering of Pruning Modes
> In the abstract (L30-32), we state: “Our analysis of the clusters learned by APTP reveals they are semantically meaningful.” Similarly, in the introduction (L98-99), we note: “We show that our prompt router learns to group the input prompts into semantic clusters.”
>
> We discuss these findings in Section 4.2 and provide the concepts assigned to each expert in Table 2 and Figure 3 (for CC3M). Results for the COCO dataset are in Appendix D.4.3 (E.4.3 in the revised pdf), including Table 4 and Figure 12 in the original paper (Figure 14 in the revised version). If these lines do not convey this information clearly or this is not the intention of the reviewer, we would appreciate further clarification from the reviewer so we can make the necessary amendments.
>
> ## 5. Limited Setting and Empirical Results
>
> We apologize for any lack of clarity regarding Table 1. The latency reductions reported are for one model evaluation, since the original Stable Diffusion model, APTP, and baselines evaluated using the same number of sampling steps. We specifically target the architectural efficiency of diffusion models. Reducing the model’s latency and memory requirements by approximately 25% on average is significant. That means that the latency of every generation step is reduced by 25%. Importantly, APTP can be combined with step-skipping, step-distillation, or caching methods to reduce the number of evaluations, leading to much greater latency reductions. As explained in **section 1.2** of the global comment, reducing the number of sampling steps is **orthogonal** to our approach.
>
> ## 6. Other Diffusion Architectures
> Please see **global response section 3**.
>
> As the reviewer requested, we also evaluated APTP on Stable Diffusion 3 (MM-DiT) model with 2 billion parameters, trained using a rectified flow objective. APTP achieved a latency reduction of 30% per sampling step, while keeping the FID/CLIP scores on MSCOCO comparable to original model's performance. These results underscore the generality of our framework across architectures. Importantly, APTP preserves the Pickscore, a proxy for human preference, on Partiprompts, an out-of-distribution benchmark, within 98% of the original model.
>
> ---
> Once again we thank the reviewer for their feedback.

---

> > ### Comment · Reviewer_vGxh · 2024-11-24
> > **Response to Authors Rebuttal**
> >
> > Thank you for your response. While some of my minor concerns have been addressed, my primary concern remains: lack of comparisons to fewer-step diffusion models.
> >
> > I understand the authors claim that pruning being orthogonal to this kind of methods. However, I believe this claim should be supported by an experiment, as this combination is non-trivial.
> >
> > Specifically, the 2 following experiments would provide valuable insights:
> > 1) Testing APTP as is, but with less diffusion steps at inference time (comparing FID, CLIP score and latency).
> > 2) More importantly, combining APTP with a step-distillation approach, showing it is indeed orthogonal (comparing both FID, CLIP score and latency).
> >
> > If the authors could provide these experiments, I would be willing to reconsider my score.
> >
> > Additionally, could the authors please upload the captions used to create Figure 7?

---

> ### Author Response · Authors · 2024-11-25
>
> Thank you for your thoughtful response and for clarifying the experiments that could address your concerns. Below, we provide the requested results and additional clarifications.
>
> ## 1. Testing APTP with Fewer Sampling Steps
> We conducted experiments to test APTP with fewer diffusion steps during inference, as requested. The results are as follows. We see that APTP can be combined with fewer sampling steps for more efficiency gains, while keeping FID, CLIP score and CMMD close the original model.
>
> | Model                          | Sampling Steps | Latency (s) | FID   | CLIP Score | CMMD  |
> |--------------------------------|----------------|-------------|-------|------------|-------|
> | APTP-COCO-Base                 | 25             | 3.1         | 22.60 | 31.32      | 0.569 |
> | APTP-COCO-Base                 | 20             | 2.6         | 24.03 | 30.65      | 0.601 |
> | APTP-COCO-Base                 | 15             | 2.1         | 25.39 | 30.34      | 0.660 |
>
> ## 2. Step-Distillation Methods
> To address the reviewer's concern, we performed **consistency distillation** using the official code from diffusers ([link](https://github.com/huggingface/diffusers/tree/main/examples/consistency_distillation)). We trained the APTP-COCO-Base model for only 2000 distillation iterations on COCO. While longer training  and a larger dataset improves results significantly, the following findings already demonstrate that APTP is orthogonal to step-distillation methods:
>
> | Model                                       | Sampling Steps | Latency (s) | FID    | CLIP Score | CMMD  |
> |--------------------------------------------|----------------|-------------|--------|------------|-------|
> | APTP-COCO-Base                              | 4              | 0.7         | 92.04  | 25.95      | 2.369 |
> | APTP-COCO-Base + Consistency Distillation   | 4              | 0.7         | 33.22  | 29.39      | 0.838 |
>
> We wanted to include experiments with the specific works mentioned by the reviewer; however, we encountered the following issues:
>
> 1. **InstaFlow**: Only pre-trained models and testing code are available; no training code is provided. This limitation is noted in their repository: [Issue #28](https://github.com/gnobitab/InstaFlow/issues/28).
> 2. **Adversarial Diffusion Distillation**: Only the model (SDXL-turbo) and inference code are provided by Stability Ai. While there is an unofficial implementation, it deviates from the original method, so we opted not to use it.
> 3. **Distillation of Guided Diffusion Models**: The code released by Google is not in PyTorch. As our implementation is based on PyTorch and diffusers, we could not integrate it in this short period.
>
> ### 3. Captions for Figure 7
> The captions used to create Figure 7 are included in **Table 8** at the end of the appendix (page 31). We recognize that this reference was not included in the figure caption. Thank you for highlighting this oversight; we will revise the figure caption in the next version of the paper.
>
> We appreciate your detailed feedback and hope this additional information addresses your concerns and encourages you to reconsider your score.

---

> > ### Comment · Reviewer_vGxh · 2024-11-25
> > **Response to Authors**
> >
> > Thank you for your response. However, I believe both of these experiments do not show orthogonality yet.
> >
> > They are missing the most important comparison: the same results **without** APTP.
> >
> > To truly understand whether APTP improves latency without affecting performance, we should test the same experiments without it, meaning 1) sampling from the diffusion model without APTP using less steps, and 2) Consistency Distillation for the original diffusion model, before APTP tuning.
> >
> > These comparisons are essential to determine whether APTP adds significant value without compromising performance.

---

> ### Author Response · Authors · 2024-11-26
>
> Thank you for your continued feedback and for highlighting the importance of including additional baselines to further support our claims. Below, we provide a detailed response to your concerns:
>
> ## 1. Previous Evidence Supporting APTP's Value
> In our paper, we have shown that APTP adds significant value to the baseline diffusion model (SD2.1). Specifically, we demonstrated that APTP as an **architecture efficiency** method reduces **memory usage** and latency while maintaining similar performance when compared to SD2.1 with the same number of sampling steps. These results are crucial as they establish that APTP enhances the baseline diffusion model independently.
>
> ## 2. Purpose of Rebuttal Experiments
> The requested experiments during  the rebuttal aimed to demonstrate that APTP is orthogonal to fewer-step sampling and step-distillation methods—showing compatibility and complementary value. We showed that as well.
>
> However, we understand your concern and agree that including results without APTP is necessary to isolate and quantify APTP's contributions within these specific contexts.
>
> ## 3. Additional Experiments
> To address your request, we  conducted the same experiments without APTP. Below, we include the updated tables with the results of the baseline (SD2.1).  Once again, we observe that with **approximately a 25% reduction in memory usage compared to SD2.1, APTP delivers comparable performance**. With a similar latency of 3.1 seconds, APTP surpasses SD2.1 in both CLIP score and CMMD (Row 1 and Row 5 of Tab. 1).
> Moreover, as shown in Tab. 2, **"APTP+consistency distillation" outperforms "SD2.1 + consistency distillation"** in all three metrics.
>
> ### Tab. 1: APTP and SD 2.1 Results with Fewer Sampling Steps
>
> | Model                          | Sampling Steps | Latency (s) | FID   | CLIP Score | CMMD  |
> |--------------------------------|----------------|-------------|-------|------------|-------|
> | APTP-COCO-Base                 | 25             | 3.1         | 22.60 | 31.32      | 0.569 |
> | APTP-COCO-Base                 | 20             | 2.6         | 24.03 | 30.65      | 0.601 |
> | APTP-COCO-Base                 | 15             | 2.1         | 25.39 | 30.34      | 0.660 |
> | SD2.1 (Baseline)               | 25             |   4.0          |   15.47    |     31.33       |   0.500    |
> | SD2.1 (Baseline)               | 20             |    3.1         |    21.80   |     31.30     |     0.598  |
> | SD2.1 (Baseline)               | 15             |    2.7         |   22.52    |     31.29       |   0.602    |
>
> ### Tab.2: Consistency Distillation Results
>
> | Model                                       | Sampling Steps | Latency (s) | FID    | CLIP Score | CMMD  |
> |--------------------------------------------|----------------|-------------|--------|------------|-------|
> | APTP-COCO-Base                              | 4              | 0.7         | 92.04  | 25.95      | 2.369 |
> | APTP-COCO-Base + Consistency Distillation   | 4              | 0.7         | 33.22  | 29.39      | 0.838 |
> | SD2.1                          | 4              |     1.0        |   78.42     |   27.76         | 1.763      |
> | SD2.1 + Consistency Distillation | 4              |  1.0           |   37.30     |  28.86          |   0.841    |
>
> ## 4. Closing Remarks
>  The additional experiments provide a more comprehensive understanding of APTP’s contributions and further reinforce our claim of orthogonality.
>
> We appreciate your constructive feedback and hope this resolves your concerns.

---

> > ### Comment · Reviewer_vGxh · 2024-11-26
> >
> > Thank you for putting the effort to answer my comments. Some of my concerns have been addressed, and so I updated my score to 6.

---

> > > ### Author Response · Authors · 2024-11-26
> > >
> > > Thank you for taking the time to review our responses and for updating your score. Your feedback has been invaluable in improving the clarity and robustness of our work.
> > >
> > > If there are any remaining concerns or additional suggestions, we would be more than happy to address them. Thank you again for your constructive engagement with our submission.

---

### Official Review · Reviewer_uiaa · 2024-11-03

**Soundness:** 2
**Presentation:** 2
**Contribution:** 2
**Rating:** 5
**Confidence:** 3

**Summary:**

The authors propose prompt-based tuning of text-to-image diffusion models, in which different sub-networks within pre-trained models are trained for different prompts/concepts. They authors performe experiments on multiple datasets to show that for a given latency, their model performs comparably to higher latency pretrained models.

**Strengths:**

1. The idea behind the paper is technically sound and novel.
2. The paper is well written.
3. The authors present some interesting interepretability experiments on expert assignment that aid the proposed concepts.

**Weaknesses:**

1. Limited experiments - Although I find the proposed ideas novel, I believe that the paper lacks extensive experimentation on
 - different types of architecture - It is currently unknown if the proposed methods is generalizable across architectures (DiT, MMDiT etc).
 - Small datasets - How does the method perform when data is limited?
 - Fine grained concepts - How does their method handle expert assignment when concepts are fine-grained (breeds of different animals)
2. Comparison to Mixture-of-Experts (MoE) models - How does the proposed method compare to other prompt-conditinal architectures like MoE text-to-image diffusion models? Currently the competitors in Table 1 (a and b) are static structural pruning baselines, but I believe the paper's contribution is prompt-conditional pruning, which demands comparison to prompt-conditional efficient architectures like MoEs like [1].
3. I am concerned about the 4 and 7 point drop in FID of the proposed method in Table 1. The authors have not presented any trade-off between latency and performance, which would help understand how limiting computational budget affects performance


[1] RAPHAEL: Text-to-Image Generation via Large Mixture of Diffusion Paths, Xue et al

**Questions:**

See Weaknesses

---

> ### Author Response · Authors · 2024-11-21
> **Response to Reviewer uiaa**
>
> We thank Reviewer uiaa for their constructive feedback as well as recognizing the novelty and soundness of our approach and the clarity of our presentation. We addressed some common concerns in the general comment. Below, we address the specific concerns raised in their review.
>
> ## 1. Generalizability Across Architectures
>
> Please see **general comment section 3.**
>
> As the reviewer requested, we evaluated APTP on Stable Diffusion 3 (MM-DiT) model with 2 billion parameters, trained using a rectified flow objective. APTP achieved a latency reduction of $30\$% per sampling step, while keeping the FID/CLIP scores on MSCOCO comparable to original model's performance. These results underscore the generality of our framework across architectures. Importantly, APTP preserves the Pickscore, a proxy for human preference, on Partiprompts, an out-of-distribution benchmark, within $98$% of the original model.
>
> ## 2. Experiments on Small and Large Datasets
>
> We acknowledge the importance of testing across varying dataset sizes. Accordingly, we evaluated APTP on two distinct scales:
>   - MSCOCO (small, ~80k images with five similar prompts per image)
>   - CC3M (larger, 3M images)
>
> MSCOCO is already small compared to datasets typically used for training T2I diffusion models. Our results demonstrate that APTP adapts effectively to different data scales, with significant improvements in convergence speed compared to baselines. This makes APTP particularly advantageous for resource-constrained settings.
>
> ## 3. Handling Fine-Grained Concepts
> To the best of our knowledge, no text-to-image datasets offer the fine-grained labeling suggested by the reviewer, such as distinguishing animal breeds while being of a sufficient length. For example, while the captioned version of the CUB dataset contains bird captions, these lack detailed subcategories and are similar in granularity to COCO captions.
>
> Handling fine-grained captions is primarily the responsibility of the router model. In our experiments, we used a sentence transformer model for its strong contrastive capabilities and efficiency on the datasets we employed. However, our framework is not tied to any specific router model. More capable router models could be employed to handle finer-grained captions and route them to specialized experts.
>
> Exploring datasets designed for fine-grained text-to-image alignment and advanced router models represents a promising direction for future work.
>
> ## 4. Comparisons to Mixture-of-Experts (MoE) Models
> We appreciate the suggestion to compare APTP with prompt-conditional MoE architectures like RAPHAEL [1]. We have included it in our related work. However, there are significant differences in scope and experimental setup:
>    1. **Resource Gap:** RAPHAEL was trained using 1,000 A100 GPUs over two months, leveraging a significantly larger computational budget. In contrast, our framework uses off-the-shelf pretrained models and requires only ~40 GPU-hours for pruning and ~120 GPU-hours for fine-tuning.
>
>    2. **Objective Difference:** RAPHAEL focuses on designing and training a new MoE-based architecture from scratch, whereas APTP focuses on prompt-based pruning of existing pretrained models. This makes APTP more resource-efficient and broadly applicable. The goal of our work is to train smaller diffusion models quickly and efficiently, which differs from approaches requiring large-scale training from scratch.
>
> Given these differences, RAPHAEL is not directly comparable to our method, and we have chosen not to include it in our comparisons.
>
> ## 5. Trade-offs Between Latency and Performance
>
> Regarding the trade-off betwen performance and latency, we pruned Stable Diffusion to two different target budgets for each dataset, with results provided in Table 1. As with any architectural pruning method, there is an inherent trade-off where performance decreases with higher sparsity levels. The results for the small and base settings of APTP illustrate this trade-off, underscoring the flexibility of our approach.
>
> APTP achieves a smaller FID drop relative to its latency gains compared to baselines. This gap narrows further with additional fine-tuning iterations. The values reported in Table 1 are based on 30k iterations using datasets of 100k and 3M samples. We expect that increasing the number of iterations and training samples will further minimize this gap.
>
> Moreover, FID is known to be an unreliable performance metric due to its sensitivity to generative artifacts [2, 3].  For a more robust evaluation, we report CLIP Score and CMMD, which better align with human-perceived quality. Our results show that APTP achieves performance levels close to the original model in these metrics.
>
> Additionally, we now include PickScore, a metric trained to reflect human preferences (**please check section (2) in the general comment**). On benchmarks like PartiPrompts, our pruned models achieves 99% of the score of SD 2.1.

---

> > ### Author Response · Authors · 2024-11-21
> >
> > ----
> > Once again we thank the reviewer for their valuable feedback, which has helped improve the quality and scope of our work.
> >
> > References:
> >
> > [1] RAPHAEL: Text-to-Image Generation via Large Mixture of Diffusion Paths, Xue et al.
> >
> > [2] Rethinking FID: Towards a Better Evaluation Metric for Image Generation, Jayasumana et al.
> >
> > [3] Pick-a-Pic: An Open Dataset of User Preferences for Text-to-Image Generation, Kirstain et al.

---

> ### Author Response · Authors · 2024-11-25
> **More results on Combining APTP with Step Distillation Methods**
>
> Dear reviewer,
>
> We now have added experiments showing that APTP is orthogonal to step reduction/distillation methods. We hope these new results will encourage you to consider increasing your score.
>
> ## 1. Testing APTP with Fewer Sampling Steps
> We conducted experiments to test APTP with fewer diffusion steps during inference, as requested. The results are as follows. We see that APTP can be combined with fewer sampling steps for more efficiency gains, while keeping FID, CLIP score and CMMD close the original model.
>
> | Model                          | Sampling Steps | Latency (s) | FID   | CLIP Score | CMMD  |
> |--------------------------------|----------------|-------------|-------|------------|-------|
> | APTP-COCO-Base                 | 25             | 3.1         | 22.60 | 31.32      | 0.569 |
> | APTP-COCO-Base                 | 20             | 2.6         | 24.03 | 30.65      | 0.601 |
> | APTP-COCO-Base                 | 15             | 2.1         | 25.39 | 30.34      | 0.660 |
> | SD2.1 (Baseline)               | 25             |   4.0          |   15.47    |     31.33       |   0.500    |
> | SD2.1 (Baseline)               | 20             |    3.1         |    21.80   |     31.30     |     0.598  |
> | SD2.1 (Baseline)               | 15             |    2.7         |   22.52    |     31.29       |   0.602    |
>
> ## 2. Step-Distillation Methods
> To address the reviewer's concern, we performed **consistency distillation** using the official code from diffusers ([link](https://github.com/huggingface/diffusers/tree/main/examples/consistency_distillation)). We trained the APTP-COCO-Base model for only 2000 distillation iterations on COCO. While longer training  and a larger dataset improves results significantly, the following findings already demonstrate that APTP is orthogonal to step-distillation methods:
>
> | Model                                       | Sampling Steps | Latency (s) | FID    | CLIP Score | CMMD  |
> |--------------------------------------------|----------------|-------------|--------|------------|-------|
> | APTP-COCO-Base                              | 4              | 0.7         | 92.04  | 25.95      | 2.369 |
> | APTP-COCO-Base + Consistency Distillation   | 4              | 0.7         | 33.22  | 29.39      | 0.838 |
> | SD2.1                          | 4              |     1.0        |   78.42     |   27.76         | 1.763      |
> | SD2.1 + Consistency Distillation | 4              |  1.0           |   37.30     |  28.86          |   0.841    |
>
> We wanted to include experiments with the specific works mentioned by the reviewers; however, we encountered the following issues:
>
> 1. **InstaFlow**: Only pre-trained models and testing code are available; no training code is provided. This limitation is noted in their repository: [Issue #28](https://github.com/gnobitab/InstaFlow/issues/28).
> 2. **Adversarial Diffusion Distillation**: Only the model (SDXL-turbo) and inference code are provided by Stability Ai. While there is an unofficial implementation, it deviates from the original method, so we opted not to use it.
> 3. **Distillation of Guided Diffusion Models**: The code released by Google is not in PyTorch. As our implementation is based on PyTorch and diffusers, we could not integrate it in this short period.

---

> ### Author Response · Authors · 2024-12-02
> **A gentle reminder**
>
> We greatly appreciate the time and effort you have invested in providing feedback. Based on your and another reviewer's suggestions, we have included comparative results with SOTA methods, which will be provided in the supplementary material of the final version.
>
> Conducting systematic, controlled evaluations of large-scale text-to-image models remains challenging, as most existing models, datasets, or implementations are not publicly available. Training a new model from scratch is prohibitively expensive, even when training code is accessible. Extending the design decision of architecture design methods to other compute budgets or settings is highly non-trivial. In contrast, our approach can be applied in a plug-and-play manner to any target model capacity, delivering competitive performance with drastically reduced hardware compute time.
>
> We compare our model to recent text-to-image models, including RAPHAEL, based on available information and their reported values, acknowledging significant differences in training datasets, iteration counts, batch sizes, and model sizes. We also include a column detailing the compute used to train these models, with estimates derived from the papers and public sources. Our findings demonstrate that APTP achieves competitive results post-training while requiring several orders of magnitude less compute time.
>
> ### Tab. 1: Comparison of APTP and SOTA Text-to-Image architecture design methods.
> | Models         |  Type  | Sampling   | #Steps | FID-30K↓ | CLIP↑  | #Params (B) | #Images (B) | Compute (GPU/TPU days)|
> |----------------|------------|------------|--------|----------|--------|-------------|-----------|-----------|
> | [GigaGAN](https://arxiv.org/abs/2303.05511)| GAN  | 1-step     | 1      | 9.09     | -      | 0.9         | 0.98      | 4783 A100|
> | [Cogview-2](https://arxiv.org/abs/2204.14217)  |AR | 1-step     | 1      | 24.0     | -      | 6.0        | 0.03      |    -      |
> | [DALL·E-2](https://arxiv.org/abs/2204.06125) |Diffusion | DDPM       | 292    | 10.39    | -      | 5.20        | 0.25      |  41667 A100     |
> | [Imagen](https://arxiv.org/abs/2205.11487)    |Diffusion| DDPM       | 256    | 7.27     | -      | 3.60        | 0.45      |   7132 TPU        |
> | [SD2.1](https://arxiv.org/abs/2112.10752)       |Diffusion| DDIM       | 50     | 9.62     | 0.304  | 0.86        | >2      |    8334 A100         |
> | [PIXART-α](https://arxiv.org/pdf/2310.00426)   |Diffusion| DPM        | 20     | 10.65    | -      | 0.6         | 0.025     |   753 A100         |
> | [SnapFusion](https://arxiv.org/pdf/2306.00980)|Diffusion | Distilled  | 8      | 13.5     | 0.308  | 0.85        | -         | >128   A100       |
> | [MobileDiffusion](https://arxiv.org/abs/2311.16567) |Diffusion| DDIM       | 50     | 8.65     | 0.325  | 0.39        | 0.15      |   7680  TPU   |
> | [RAPHAEL](https://arxiv.org/abs/2305.18295)|Diffusion| DDIM       | -     | 6.61     | -  |3.0       | >5     | 60000    A100   |
> | APTP-Base (@30k)|Diffusion | DDIM       | 50     |19.14     | 0.318  | 0.65   | 0.0001      | 6.5  A100     |
> ----
>
>
> As the discussion period ends today, we would like to address any remaining concerns you may have. We kindly request that you review the new results, the revised submission and our rebuttal and consider providing additional feedback. If you find the improvements satisfactory, we would be especially grateful if you might consider adjusting your score. Thank you once again for your valuable input and for helping to enhance the quality of our work.

---

### Official Review · Reviewer_Ff7o · 2024-11-04

**Soundness:** 2
**Presentation:** 2
**Contribution:** 2
**Rating:** 6
**Confidence:** 2

**Summary:**

The paper introduces adaptive prompt-based pruning strategy to reduce the computation cost of diffusion model. The proposed approach involves encoding input prompts into architecture embeddings, which are mapped to specialized architecture codes. These codes determine the routing of each prompt to a pruned sub-network. By training a prompt router using a combination of contrastive learning and optimal transport, the proposed method ensures that prompts are dynamically assigned to appropriate sub-networks. The results of the paper demonstrate the reduction in computational cost while maintaining FID and CLIP scores.

**Strengths:**

1. The paper introduces a novel approach by proposing adaptive prompt-based pruning that routes input prompts to specialized pruned sub-networks based on their characteristics. This represents a difference from conventional static and dynamic pruning methods,
2. The empirical results training on datasets like CC3M and MS-COCO demonstrate the method’s effectiveness compared to other pruning methods. The results show that the proposed method outperforms other baselines by significantly reducing computational cost while maintaining or improving output quality as measured by metrics like FID, CLIP score, and CMMD score.

**Weaknesses:**

The major concern is the empirical evaluation of the proposed method:

1. as stated in the paper, most organizations typically fine-tune pre-trained diffusion models on their target data but evaluate these models on broader benchmarks to demonstrate generalizability. In this study, however, the authors only fine-tune their model on CC3M and MS-COCO and limit their evaluation to the corresponding validation sets. Expanding the evaluation to a common benchmark would better showcase the model’s generalization capabilities. Specifically, demonstrating that the prompt router can handle prompts outside the training distribution would be more convincing.

2. The paper also references other model pruning methods, such as MobileDiffusion[1], SnapFusion[2], and LD-Pruner[3]. However, it does not include quantitative comparisons with these approaches. It would be helpful for the authors to explain why these comparisons were omitted.

3. In efficient inference for stable diffusion, recent papers show that one-step or few-step generation can speed up the generation. This paper does not include comparisons with methods like INSTAFLOW[4], which would have provided valuable insights into how APTP compares with state-of-the-art approaches in rapid generation.



[1] Zhao, Yang, et al. "Mobilediffusion: Subsecond text-to-image generation on mobile devices." arXiv preprint arXiv:2311.16567 (2023).

[2] Li, Yanyu, et al. "Snapfusion: Text-to-image diffusion model on mobile devices within two seconds." Advances in Neural Information Processing Systems 36 (2024).

[3] Castells, Thibault, et al. "LD-Pruner: Efficient Pruning of Latent Diffusion Models using Task-Agnostic Insights." Proceedings of the IEEE/CVF Conference on Computer Vision and Pattern Recognition. 2024.

[4] Liu, Xingchao, et al. "Instaflow: One step is enough for high-quality diffusion-based text-to-image generation." The Twelfth International Conference on Learning Representations. 2023.

**Questions:**

1. The authors utilize a pre-trained sentence transformer as the prompt encoder in their training process. Do the authors have any insights into how the size of the prompt encoder influences the overall performance, as the size of the prompt encoder will affect the models' ability to understand the input prompt?

2. Training diffusion models often incorporate classifier-free guidance. Is the proposed method compatible with training under this manner?

---

> ### Author Response · Authors · 2024-11-21
> **Response to Reviewer Ff7o**
>
> We thank Reviewer Ff7o for their detailed feedback. We are pleased that the reviewer found our method novel. While we addressed most of the reviewer's concerns in the general comment, we reiterate them here and respond to additional specific points.
>
> ## 1. Expanding Evaluation to a Common Benchmark
> Please **section (2)** of the general comment where we explain that we evaluated the APTP-pruned model on the Partiprompt dataset using the  PickScore, which uses a model trained to mimic human preferences for synthetically generated images.
>
> ## 2. Comparison with MobileDiffusion [1], SnapFusion [2], and LD-Pruner [3]
> Please refer to point **(1.1)** in the general comment for a detailed discussion regarding MobileDiffusion and SnapFusion.
>
> We thank the reviewer for bringing LD-Pruner to our attention. We have now included it in the related work section. Unfortunately, its implementation code has not been released, making direct comparisons infeasible within the short discussion period.
>
>
> ## 3. Comparison with one/few-step generation methods
>
> Please refer to **section (1.2)** in the general comment.
>
> Our method, as an architecture efficiency method, is **orthogonal** to  methods that reduce the number of evaluations (e.g., by decreasing sampling steps) for more efficiency gains. It can be combined with them. As an example, the **InstaFlow method** suggested by the reviewer (which we have now included in the related work) could be applied during the fine-tuning phase of pruned models. Currently, we use a combination of DDPM and distillation loss, but following InstaFlow's recipe, a pruned Stable Diffusion model could be converted to a Rectified Flow model capable of generating images in fewer steps.
>
> ## 4. Influence of the Prompt Encoder's size
>
> Please see **section (4) in the general comment** for detailed explanations.
>
> To clarify, the text encoder of the diffusion model (e.g., CLIP) remains unchanged in our experiments. We use a sentence transformer as the router module to assign prompts to architecture codes based on their complexity. This model, with $\sim$105M parameters, effectively handled the datasets and pruning budgets in our experiments.
>
> As the prompts used in our experiments are relatively short, a sentence transformer model seemed sufficient. However, APTP is a flexible framework, and organizations could replace the sentence transformer with larger models, such as the T5 text encoder or even a larger LLM, to handle more complex or longer prompts. To maintain academic focus and prioritize efficiency, we chose the sentence transformer model for our study.
>
> ## 5. CFG Compatibility:
>
> Yes, our method is fully compatible with CFG. In fact, CFG was used in our experiments with a scale of $7.5$ (L1047) to generate samples. More broadly, *any* training technique for diffusion models can be applied during the fine-tuning phase of pruned models without modification.
>
> ---
> We thank the reviewer again for their feedback on our work. We hope we have addressed their concerns adequately.
>
> References:
>
> [1]: Zhao, Yang, et al. "MobileDiffusion: Subsecond text-to-image generation on mobile devices." arXiv preprint arXiv:2311.16567 (2023).
>
> [2]: Li, Yanyu, et al. "Snapfusion: Text-to-image diffusion model on mobile devices within two seconds." Advances in Neural Information Processing Systems 36 (2024).
>
> [3]: Castells, Thibault et, all. "LD-Pruner: Efficient Pruning of Latent Diffusion Models using Task-Agnostic Insights" arXiv preprint arXiv:2404.11936 (2024)

---

> ### Author Response · Authors · 2024-11-25
> **Results for Combining APTP with Step Distillation Methods**
>
> Dear reviewer,
>
> We now have added experiments showing that APTP is orthogonal to step reduction/distillation methods. We hope these new results will encourage you to consider increasing your score.
>
> ## 1. Testing APTP with Fewer Sampling Steps
> We conducted experiments to test APTP with fewer diffusion steps during inference, as requested. The results are as follows. We see that APTP can be combined with fewer sampling steps for more efficiency gains, while keeping FID, CLIP score and CMMD close the original model.
>
> | Model                          | Sampling Steps | Latency (s) | FID   | CLIP Score | CMMD  |
> |--------------------------------|----------------|-------------|-------|------------|-------|
> | APTP-COCO-Base                 | 25             | 3.1         | 22.60 | 31.32      | 0.569 |
> | APTP-COCO-Base                 | 20             | 2.6         | 24.03 | 30.65      | 0.601 |
> | APTP-COCO-Base                 | 15             | 2.1         | 25.39 | 30.34      | 0.660 |
>
> ## 2. Step-Distillation Methods
> To address the reviewer's concern, we performed **consistency distillation** using the official code from diffusers ([link](https://github.com/huggingface/diffusers/tree/main/examples/consistency_distillation)). We trained the APTP-COCO-Base model for only 2000 distillation iterations on COCO training data. While longer training  and a larger dataset improves results significantly, the following findings already demonstrate that APTP is orthogonal to step-distillation methods:
>
> | Model                                       | Sampling Steps | Latency (s) | FID    | CLIP Score | CMMD  |
> |--------------------------------------------|----------------|-------------|--------|------------|-------|
> | APTP-COCO-Base                              | 4              | 0.7         | 92.04  | 25.95      | 2.369 |
> | APTP-COCO-Base + Consistency Distillation   | 4              | 0.7         | 33.22  | 29.39      | 0.838 |
>
> We wanted to include experiments with the specific works mentioned by the reviewers; however, we encountered the following issues:
>
> 1. **InstaFlow**: Only pre-trained models and testing code are available; no training code is provided. This limitation is noted in their repository: [Issue #28](https://github.com/gnobitab/InstaFlow/issues/28).
> 2. **Adversarial Diffusion Distillation**: Only the model (SDXL-turbo) and inference code are provided by Stability Ai. While there is an unofficial implementation, it deviates from the original method, so we opted not to use it.
> 3. **Distillation of Guided Diffusion Models**: The code released by Google is not in PyTorch. As our implementation is based on PyTorch and diffusers, we could not integrate it in this short period.

---

> ### Comment · Reviewer_Ff7o · 2024-11-26
>
> Thank you for your rebuttal. I appreciate the effort in addressing my concerns, and I believe most of them have been adequately resolved. However, I still find it crucial to include a comparison with state-of-the-art (SOTA) methods, particularly those focusing on architecture design that closely aligns with your work. While it is understandable if your method performs worse than some SOTA approaches due to differences in training dataset size or training time, it is essential to quantify the performance gap between your method and these approaches training from scratch. For example, in MobileDiffusion, Table 2 highlights methods that outperform theirs but provides context by detailing the amount of data and model size, effectively demonstrating their method's efficiency despite these limitations. Including a similar comparison in your work would significantly enhance its contribution.

---

> ### Author Response · Authors · 2024-11-26
>
> We sincerely thank the reviewer for their detailed feedback and valuable suggestions. As recommended, we have included comparative results with SOTA methods, which will be provided in the supplementary material of the final version.
>
> Conducting systematic, controlled evaluations of large-scale text-to-image models remains challenging, as most existing models, datasets, or implementations are not publicly available. Training a new model from scratch is prohibitively expensive, even when training code is accessible. Extending the design decision of architecture design methods to other compute budgets or settings is highly non-trivial. In contrast, our approach can be applied in a plug-and-play manner to any target model capacity, delivering competitive performance with drastically reduced hardware compute time.
>
> We compare our model to recent text-to-image models based on available information and their reported values, acknowledging significant differences in training datasets, iteration counts, batch sizes, and model sizes. We also include a column detailing the compute used to train these models, with estimates derived from the papers and public sources. Our findings demonstrate that APTP achieves competitive results post-training while requiring several orders of magnitude less compute time.
>
> ### Tab. 1: Comparison of APTP and SOTA Text-to-Image architecture design methods.
> | Models         |  Type  | Sampling   | #Steps | FID-30K↓ | CLIP↑  | #Params (B) | #Images (B) | Compute (GPU/TPU days)|
> |----------------|------------|------------|--------|----------|--------|-------------|-----------|-----------|
> | [GigaGAN](https://arxiv.org/abs/2303.05511)| GAN  | 1-step     | 1      | 9.09     | -      | 0.9         | 0.98      | 4783 A100|
> | [Cogview-2](https://arxiv.org/abs/2204.14217)  |AR | 1-step     | 1      | 24.0     | -      | 6.0        | 0.03      |    -      |
> | [DALL·E-2](https://arxiv.org/abs/2204.06125) |Diffusion | DDPM       | 292    | 10.39    | -      | 5.20        | 0.25      |  41667 A100     |
> | [Imagen](https://arxiv.org/abs/2205.11487)    |Diffusion| DDPM       | 256    | 7.27     | -      | 3.60        | 0.45      |   7132 TPU        |
> | [SD2.1](https://arxiv.org/abs/2112.10752)       |Diffusion| DDIM       | 50     | 9.62     | 0.304  | 0.86        | >2      |    8334 A100         |
> | [PIXART-α](https://arxiv.org/pdf/2310.00426)   |Diffusion| DPM        | 20     | 10.65    | -      | 0.6         | 0.025     |   753 A100         |
> | [SnapFusion](https://arxiv.org/pdf/2306.00980)|Diffusion | Distilled  | 8      | 13.5     | 0.308  | 0.85        | -         | >128   A100       |
> | [MobileDiffusion](https://arxiv.org/abs/2311.16567) |Diffusion| DDIM       | 50     | 8.65     | 0.325  | 0.39        | 0.15      |   7680  TPU   |
> | [RAPHAEL](https://arxiv.org/abs/2305.18295)|Diffusion| DDIM       | -     | 6.61     | -  |3.0       | >5     | 60000    A100   |
> | APTP-Base (@30k)|Diffusion | DDIM       | 50     |19.14     | 0.318  | 0.65   | 0.0001      | 6.5  A100     |
> ----
>
> We hope these results clarify any remaining questions and address concerns.

---

> > ### Comment · Reviewer_Ff7o · 2024-11-27
> >
> > Thank you for your detailed rebuttal. I appreciate the effort you put into addressing my concerns. Most of my concerns have been resolved. I will maintain my score and lean towards a positive evaluation of the paper.

---

> > > ### Author Response · Authors · 2024-11-27
> > >
> > > Thank you for your thoughtful feedback and for taking the time to review our responses. We are glad to hear that most of your concerns have been resolved. Your constructive evaluation is greatly appreciated,

---

### Author Response · Authors · 2024-11-21
**Global Response to All Reviewers (1/3)**

We thank the reviewers for their efforts and valuable feedback. We address the common concerns among them here.

## 1. Comparison with related works [1, 2, 3, 4, 5, 6]) (R-Ff7o, R-uiaa, R-vGxh):
We describe that the scope of our method is different from the works that the reviewers mentioned in the following:

### **1-1. Comparison with MobileDiffusion [1] (Architecture Design) and  SnapFusion[2] (Architecture Search) methods (R-Ff7o) :**
We did not compare with MobileDiffusion and SnapFusion as the scope as well as the experimental setup in our paper is significantly different from these methods.

As we describe in the Related Work section, architecture design methods like MobileDiffusion [1] develop heuristics to design an efficient architecture guided by some proxy metrics like FID and CLIP scores on MSCOCO. Similarly, SnapFusion [2] as an architecture search approach trains an architecture with elastic dimensions. Then, it searches for a performant sub-network of it while optimizing for CLIP score on MSCOCO and latency. Yet, these methods have two key drawbacks:
- They require extremely large training data size and compute budget to train their models and validate their design choices. For instance, MobileDiffusion uses a training dataset of 150 million text-image pairs from the public web and consumes approximately 512 TPUs spanning 15 days to complete the network search. In addition, SnapFusion uses internal proprietary data of unknown size to train its model with 256 NVIDIA A100 GPUs.
- Their heuristics and design choices are either non-trivial (MobileDiffusion) or costly to generalize (SnapFusion) to new compute budgets and datasets.

In contrast to these methods, APTP addresses the inference efficiency of a computationally intensive model *after* its pretraining phase by pruning it in a prompt-based manner while optimizing its performance on a *target* data distribution. We prune Stable Diffusion V2.1 using MSCOCO and CC3M, and we show the advantage of APTP compared to recent static pruning baselines for diffusion models.

In summary, our work and these methods differ in the following aspects:

   1. **Required Training Resources:** MobileDiffusion and SnapFusion train models from scratch, and their training process consumes tens of thousands of GPU/TPU hours. For instance, MobileDiffusion uses 15 Days $\times$ 24 hours $\times$ 512 TPUs = 184320 TPU hours. In contrast, our pruning phase requires approximately 40 GPU hours, and its fine-tuning takes around 120 GPU hours.

   2. **Training dataset size:** While MobileDiffusion and SnapFusion train on tens of millions of data points, we demonstrate that our method works effectively with merely 80k images in the MSCOCO training set.

   3. **Difference in Objectives:** Most importantly, our work aims to show that not all prompts necessitate the full capacity of a pre-trained diffusion model and that one-arch-for-all static pruning architecture approaches are suboptimal for T2I models. We propose a fast and efficient framework to prune an existing off-the-shelf pre-trained T2I diffusion model into a Mixture of Experts, enabling dynamic budget adjustment based on the prompt's complexity. Unlike designing a new model like MobileDiffusion or searching for it from scratch like SnapFusion, our framework is applicable to *any* pretrained diffusion model.

### **1-2. Comparison with Few/One-Step Generation using distillation [3,4,5] and Caching [6] methods(R-Ff7o, R-uiaa, R-vGxh)**
We discussed in the introduction (L47-51) and related work (L925-930 in the revised pdf) sections that step-reduction methods, like few-step generation ones [3, 4, 5] and caching [6], are **orthogonal** to architectural efficiency methods like ours. In more details, these methods address the "high number of sampling steps" aspect of the broader challenge of the "slow sampling process of diffusion models." They are complementary rather than mutually exclusive with "architectural efficiency of diffusion models." In fact, one can combine step-reduction methods with our approach to further improve inference efficiency, highlighting that our method is not an alternative but a complementary solution to these techniques. Accordingly, the scope of our paper is the architectural efficiency of diffusion models, and we benchmarked APTP against the recent pruning techniques to validate the effectiveness of APTP.

We emphasize that the latency reduction of our method is a result of **model size and memory usage reduction**, which is the **goal of architectural efficiency approaches**. The benefit of architectural efficiency methods is not limited to improved latency. **For instance, one cannot deploy a diffusion model that requires 40GB GPU memory on a GPU with 24GB of memory, regardless of the number of sampling steps and sampling speed-up techniques like [3, 4, 5, 6] that they employ.** In contrast, they can prune it using APTP to smaller experts and deploy an expert on it.

---

> ### Author Response · Authors · 2024-11-21
> **Global Response to All Reviewers (2/3)**
>
> ## 2. Generalization of the pruned models to out-of-distribution prompts (R-Ff7o, R-vGxh, R-xWxf)
>
> We evaluate the Stable Diffusion 2.1, APTP, and the baselines on the **Partiprompts** [9] dataset as suggested by Reviewer R-xWxf. The prompts in this benchmark can be considered out-of-distribution for our model as many of them are significantly longer and semantically different from the ones in MSCOCO and CC3M. We report the **PickScore** [10] as a **proxy for human preference** and present the results in Table 2.a and 2.b below. We can observe that the Pickscore of the pruned model is only $1$% below the original Stable Diffusion 2.1, indicating that the pruned model can preserve the general knowledge and generation capabilities of the Stable Diffusion model.
>
> ### Table2: Results on PartiPrompts
> We report performance metrics using samples generated at the resolution of 768.
> We measure models' MACs/Latency with the input resolution of 768 on an A100 GPU.
> @30/50k shows fine-tuning iterations after pruning.
>
> #### 2.a Train on CC3M
> | Method                                        | MACs (@768) | Latency ($\downarrow$) (Sec/Sample) (@768) | PickScore ($\uparrow$) |
> |-----------------------------------------------|-------------|--------------------------------------------|------------------------|
> | Norm @50k             | 1185.3G     | 3.4                                        | 18.114                 |
> | SP @30k    | 1192.1G     | 3.5                                        | 18.727                 |
> | BKSDM @30k                    | 1180.0G     | 3.3                                        | 19.491                 |
> | APTP(0.66) @30k                           | 916.3G      | 2.6                                        | 19.597                 |
> | APTP(0.85) @30k                               | 1182.8G     | 3.4                                        | 21.049                 |
> | SD 2.1                                        | 1384.2G     | 4.0                                        | 21.316                 |
>
> #### 2.b Train on MS-COCO
> | Method                                         | MACs (@768) | Latency ($\downarrow$) (Sec/Sample) (@768) | PickScore ($\uparrow$) |
> |------------------------------------------------|-------------|--------------------------------------------|------------------------|
> | Norm @50k              | 1077.4G     | 3.1                                        | 18.563                 |
> | SP @30k     | 1071.4G     | 3.3                                        | 19.317                 |
> | BKSDM @30k                     | 1085.4G     | 3.1                                        | 19.941                 |
> | APTP(0.64) @30k                                | 890.0G      | 2.5                                        | 20.626                 |
> | APTP(0.78) @30k                                | 1076.6G     | 3.1                                        | 21.150                 |
> | SD 2.1                                         | 1384.2G     | 4.0                                        | 21.316                 |
>
> We also demonstrate the generalization of our pruned model in **artistic styles**. We refer to **Fig. 6** which we have currently put at the beginning of the appendix (page 17 of the revised pdf) for easy reference. Fig. 6 shows generations from the original Stable Diffusion model and the APTP-pruned model on both CC3M and COCO across various styles. These results, particularly the results of the model pruned on MS-COCO, highlight the ability of APTP to generalize to concepts not present in the target dataset.

---

> ### Author Response · Authors · 2024-11-21
> **Global Response to All Reviewers (3/3)**
>
> ## 3. Generalization of the proposed method to other diffusion architectures (R-uiaa, R-vGxh, R-xWxf)
> We first note that our framework, APTP, is architecture-agnostic, and none of the components of APTP are tailored to the U-Net, diffusion loss function, or sampling method used in Stable Diffusion. We utilized the Stable Diffusion model in our experiments as it enabled us to perform fair and straightforward comparisons with the pruning baselines that used it, not due to any characteristics of the U-Net used in Stable Diffusion.
>
> To verify the effectiveness of APTP on other architectures, other model sizes, and other diffusion objectives, we use it to prune the **Stable Diffusion-3-medium** model which is a 2B parameter MM-DIT model trained with the Rectified Flow objective. The results are shown in the Tab. 1 below.
>
> APTP reduces the memory usage and latency of the one forward evaluation of the model by approximately $30$% while having similar CLIP/CMMD/FID scores on the 5K prompts of COCO-2017 validation data after only 20000 fine-tuning iterations. Furthermore, the Pickscore on Partiprompts only reduces by $2$%. These results demonstrate the generality of APTP to prune different T2I diffusion models' architectures, sizes, and loss functions.
>
> ### Table 1: Stable Diffusion 3 Medium (MM-DiT) Performance Metrics: COCO-2017-Validation and PartiPrompts
>
> | Method                   | MACS                   | Latency(sec/sample)                   |FID ($\downarrow$) | CLIP Score ($\uparrow$) | CMMD ($\downarrow$) | PickScore ($\uparrow$) |
> |--------------------------|--------------------------|--------------------------|--------------------|--------------------------|---------------------|-------------------------|
> | APTP(0.7) @20k           |  3213.9G |7.1       |  36.32| 29.12                   | 0.674               | 22.057                  |
> | Stable Diffusion 3-medium |    4463.8G    | 10.0 | 32.28             | 29.31                   | 0.606               | 22.501                  |
>
>
> ## 4. Effect of the Router's Prompt Encoder size (R-Ff7o, R-xWxf):
> In our experiments, we used a sentence transformer model with $\sim105$M parameters as our router's backbone and found that it works well for different datasets and pruning budgets. Our intuition is that as the prompts in the datasets we used have relatively small lengths, one does not need a giant language model to route the input prompts to experts in APTP. Yet, our framework is flexible enough to enable organizations to employ language models with higher capacities like T5 text encoder or even larger LLMs as the router for handling significantly more complex and longer prompts
>
> As we were experimenting with an academic infrastructure and prioritized efficiency, we employed a sentence transformer model that can encode $\sim2800$ sentences per second [7], equivalent to $0.0003$ seconds latency. Therefore, it adds a negligible latency to the one for our pruned model (R-xWxf).
>
>
> ---
> Once again we thank the reviewers for their valuable feedback.
>
> References:
>
> [1]: Zhao, Yang, et al. "MobileDiffusion: Subsecond text-to-image generation on mobile devices." arXiv preprint arXiv:2311.16567 (2023).
>
> [2]: Li, Yanyu, et al. "Snapfusion: Text-to-image diffusion model on mobile devices within two seconds." Advances in Neural Information Processing Systems 36 (2024).
>
> [3]: Liu, Xingchao, et al. "Instaflow: One step is enough for high-quality diffusion-based text-to-image generation." The Twelfth International Conference on Learning Representations. 2023.
>
> [4]: Sauer, Axel, et al. "Adversarial diffusion distillation." European Conference on Computer Vision. Springer, Cham, 2025.
>
> [5]: Meng, Chenlin, et al. "On distillation of guided diffusion models." Proceedings of the IEEE/CVF Conference on Computer Vision and Pattern Recognition. 2023.
>
> [6]: Ma, Xinyin, Gongfan Fang, and Xinchao Wang. "Deepcache: Accelerating diffusion models for free." Proceedings of the IEEE/CVF Conference on Computer Vision and Pattern Recognition. 2024.
>
> [7] https://www.sbert.net/docs/sentence_transformer/pretrained_models.html#original-models
>
> [8] RAPHAEL: Text-to-Image Generation via Large Mixture of Diffusion Paths, Xue et al
>
> [9] Scaling Autoregressive Models for Content-Rich Text-to-Image Generation, Yu et al
>
> [10] Pick-a-Pic: An Open Dataset of User Preferences for Text-to-Image Generation, Kirstain et al

---

### Meta-Review · Area_Chair_TGPd · 2024-12-19

**Metareview:**

In contrast to the traditional approach where only one pruned T2I model is called by all prompts, this paper proposes to apply different prompts to different pruned models, determined by a router model. This router model is learned by contrastive learning and optimal transport.

The evaluation of this approach is limited to relatively small datasets, and it is still unknown whether the router can be generalized to outside of training distribution.

Overall, it's an interesting idea with some good results. I agree with one reviewer that "The core idea is an interesting combination of known concepts in a new, but highly relevant setup."

**Additional Comments On Reviewer Discussion:**

Further comparison with MobileDiffusion and SnapFusion have been added after rebuttal. A new evaluation on Partiprompt is added after rebuttal, etc.

Overall the rating increased to 6, 5, 6, 8 after rebuttal. I am leaning towards accepting this paper.

---

### Decision · Program_Chairs · 2025-01-22

Accept (Poster)